# Map of synthetic rescue interactions for the Fanconi anemia DNA repair pathway identifies USP48

Georgia Velimezi[1], Lydia Robinson-Garcia[1], Francisco Muñoz-Martínez [2], Wouter W. Wiegant[3], Joana Ferreira da Silva[1], Michel Owusu[1], Martin Moder[1], Marc Wiedner[1], Sara Brin Rosenthal[4], Kathleen M. Fisch[4], Jason Moffat[5], Jörg Menche [1], Haico van Attikum[3], Stephen P. Jackson [2] Joanna I. Loizou [1]

Defects in DNA repair can cause various genetic diseases with severe pathological phenotypes. Fanconi anemia (FA) is a rare disease characterized by bone marrow failure, developmental abnormalities, and increased cancer risk that is caused by defective repair of DNA interstrand crosslinks (ICLs). Here, we identify the deubiquitylating enzyme USP48 as synthetic viable for FA-gene deficiencies by performing genome-wide loss-of-function screens across a panel of human haploid isogenic FA-defective cells (FANCA, FANCC, FANCG, FANCI, FANCD2). Thus, as compared to FA-defective cells alone, FA-deficient cells additionally lacking USP48 are less sensitive to genotoxic stress induced by ICL agents and display enhanced, BRCA1-dependent, clearance of DNA damage. Consequently, USP48 inactivation reduces chromosomal instability of FA-defective cells. Our results highlight a role for USP48 in controlling DNA repair and suggest it as a potential target that could be therapeutically exploited for FA.

[1] CeMM Research Center for Molecular Medicine of the Austrian Academy of Sciences, Lazarettgasse 14, AKH BT 25.3, 1090 Vienna, Austria. [2] The Gurdon Institute and Department of Biochemistry, University of Cambridge, Tennis Court Road, Cambridge CB2 1QN, UK. [3] Department of Human Genetics, Leiden University Medical Center, Leiden, Einthovenweg 20, 2333 ZC Leiden, The Netherlands. [4] Center for Computational Biology & Bioinformatics, Department of Medicine, University of California, San Diego, 9500 Gilman Drive #0681, La Jolla, CA 92093, USA. [5] Donnelly Centre and Banting and Best Department of Medical Research, University of Toronto, Toronto, ON M5S 1A8, Canada. These authors contributed equally: Georgia Velimezi, Lydia Robinson-Garcia. Correspondence and requests for materials should be addressed to J.l.L. (email: jloizou@cemm.oeaw.ac.at)

The human genome is constantly exposed to various sources of DNA damage that can arise from either endogenous or exogenous sources. To deal with this stress, cells possess several highly conserved and effective mechanisms for DNA repair. If these repair mechanisms are defective, due to germline mutations in relevant DNA repair genes, rare diseases with DNA repair deficiencies can arise[1,2]. One such disease is Fanconi anemia (FA), which is characterized by chromosomal instability, bone marrow failure, and cancer predisposition, for which inadequate treatments are currently available[3,4]. FA is caused by mutations in genes encoding components of the FA pathway, which mediates repair of DNA interstrand crosslinks (ICLs), highly toxic lesions that block DNA replication and transcription. Consequently, cells that have disruptive mutations in FA genes exhibit increased sensitivity to DNA ICL-inducing agents[3,4].

The classical concept of synthetic viability (also termed synthetic rescue or genetic suppression), in combination with the use of advanced and high-throughput methods allows for the development of new approaches to ameliorate defects associated with human genetic diseases[5–9]. Moreover, the identification of such interactions provides significant insights into the mechanisms underlying molecular processes and crosstalk between cellular pathways. To explore, in an unbiased and systematic manner, genetic synthetic-viable interactions for FA deficiency, we have used human haploid genetic screens—a powerful approach that can identify genetic interactions in human cells[10–12]. Thus, we have used a previously described gene-trap retrovirus[10] to mutagenize a panel of human cell lines individually carrying mutations in five different FA genes (*FANCA*, *FANCC*, *FANCG*, *FANCI*, and *FANCD2*). As described herein, through this approach, we identify USP48 as a deubiquitylating enzyme (DUB) whose loss alleviates increased sensitivity and genomic instability of FA-deficient cells following DNA crosslinking damage, while enhancing the recruitment of homologous recombination (HR) markers and histone H2A ubiquitylation.

## Results

**Map of synthetic rescue interactions in FA cells**. In order to mimic the defect observed in FA in a cellular system that is appropriate for further genetic manipulations, we generated CRISPR-Cas9-mediated knockout HAP1 cells with individual mutations in five FA genes: *FANCA*, *FANCC*, *FANCG*, *FANCI*, and *FANCD2*. We selected these genes based on their clinical relevance (FANCA, FANCC, and FANCG account for the majority of mutations in FA patients) but also based on their molecular function, as FANCA, FANCC, and FANCG are part of the core complex, while FANCI and FANCD2 function downstream during ICL repair, in order to broadly cover the FA pathway[3,4]. The HAP1 cell line is near-haploid, making it ideal for genetic screens using insertion mutagenesis, since disruptive mutations in a single allele will result in a knockout[10]. We confirmed the generation of frameshift mutations and loss of protein expression in the knockout cell lines by Sanger sequencing and immunoblotting (Supplementary Fig. 1a, b). In line with a previous report, we noted that loss of FANCA affected the protein stability of FANCG and vice-versa[13]. We also observed mutual interdependences for FANCI and FANCD2 protein stabilities. The expected hypersensitivity of the FA knockout cells to the DNA crosslinking agent mitomycin C (MMC) was confirmed by cell viability assays (Fig. 1a) and clonogenic survival assays (Supplementary Fig. 1c).

To screen for potential genetic disruptions that alleviate the DNA damage hypersensitivity of FA-defective cells, we inserted, via retroviral delivery, a gene-trap cassette that contains a splice acceptor site into the HAP1 knockout cells. This method has been previously used successfully in haploid genetic screens[10–12]. We next treated the mutagenized cells with MMC at a dose that we predetermined as selectively killing the FA-deficient cells, thereby providing a selective pressure to enrich for resistant cells (Fig. 1b). Genomic DNA was then extracted and subjected to next generation sequencing (NGS), allowing for the mapping of both the position and orientation of the gene-trap cassette. As a control, we additionally analyzed a non-selected, untreated mutagenized population of wild-type (WT) HAP1 cells[14].

In order to identify cellular processes and pathways represented by the genes that specifically rescue FA-defective cells but not WT cells, we overlaid the significantly enriched hits onto a manually curated interactome network of physical protein–protein interactions[15]. We identified neighboring genes to this initial set using network propagation based on a previously described method[16]. Groups of highly interconnected genes were revealed using a modularity maximization clustering algorithm[17], which after functional enrichment analysis of each cluster, showed significant enrichment for biologically-relevant canonical pathways (Fig. 1c; Supplementary Data 1). Clusters including "Ubiquitin-mediated proteolysis" and "Pathways in cancer" demonstrate the high connectivity among the identified genes. Also displayed are the DNA repair genes that were not identified tightly connected in a cluster but rather falling within the above two mentioned clusters (Fig. 1c; Supplementary Data 1).

**USP48 loss alleviates sensitivity of FA cells to ICLs**. We next created a synthetic-viable interaction network from all the identified rescue protein-coding genes (Fig. 2a; Supplementary Data 2). Closer examination of the top 10 individual genes revealed *NQO1* and *USP48* as the most recurrently targeted and significantly enriched genes, based on *fdr*-corrected *p*-values (*q*-values) (Fig. 2b). We, therefore, chose to focus on these for our ensuing analyses. Other enriched genes, notably *LAMTOR1* and *LAMTOR5*, which are members of the LAMTOR complex and promote activation of the mTOR signaling pathway, will be the subject of our ensuing studies. In addition, we note that we have recently reported that inactivation of members of the BLM complex (BLM, TOP3A, and RMI1), which appear among the top shared hits among FA-deficient cell screens, can rescue the survival of FA-deficient cells in response to ICL induction[18].

NQO1 is a quinone reductase that has been shown to contribute to MMC bio-activation in cells, thereby allowing its toxic actions[19]. Furthermore, expression levels of NQO1 in human tumors have been positively correlated to enhanced sensitivity to MMC treatment[20]. As expected from its function in MMC activation, disruptive insertions within *NQO1* were highly significantly enriched in wild-type (WT) cells as well as FA-deficient cells selected for MMC resistance, indicating a general mode-of-action irrespective of the DNA repair status of the cell line.

More interestingly, mutagenic insertions within *USP48*, a deubiquitylating enzyme (DUB), were found as significantly enriched in resistant populations of all the FA-deficient backgrounds but not in WT cells, potentially indicating a function specifically connected to the DNA repair defect in FA cells. Moreover, mapping of the insertion sites of the gene-trap cassette in *USP48* showed that the majority of insertions were localized upstream in the gene or at a region corresponding to the catalytic domain of USP48 (Supplementary Fig. 2a), indicative of disruptive mutations resulting in loss of function. We next validated this rescue interaction by generating, via de novo CRISPR-Cas9 gene editing, a HAP1 cell line double mutant for FANCC and USP48 (Fig. 3a and Supplementary Fig. 2b). The resulting double mutant, Δ*USP48* Δ*FANCC*, was more resistant to

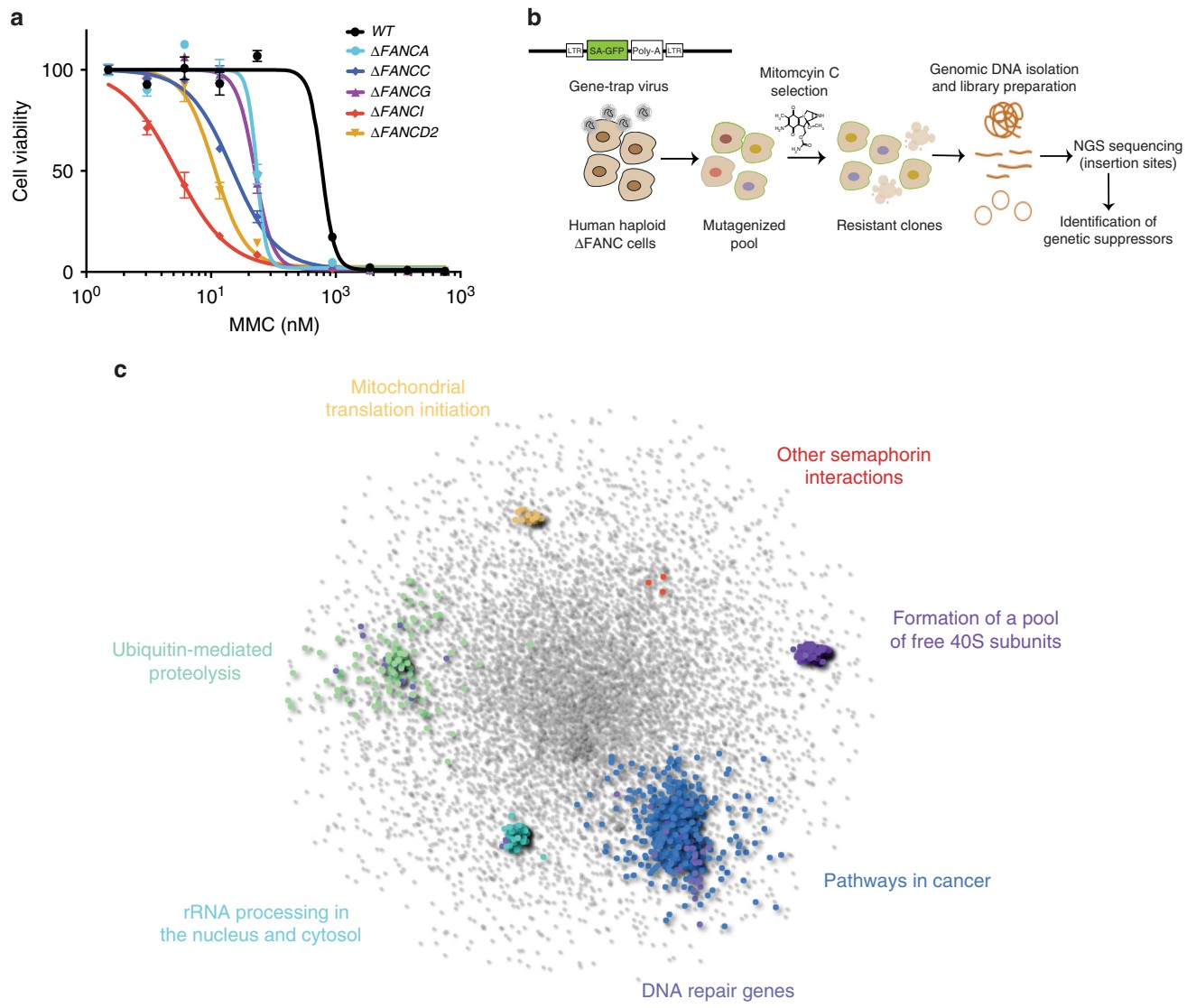

**Fig. 1** Genetic screens identify synthetic-viable interactions for FA. **a** Dose–response of FA-defective HAP1 cell lines to MMC exposure. Cells were treated for 3 days and viability was measured by Cell Titer Glo®. Error bars indicate S.E.M (standard error of the mean) of triplicates. **b** Workflow for the identification of genetic synthetic-viable interactions for FA-defective human haploid HAP1 cells following MMC exposure by gene-trap-based insertional mutagenesis. **c** Protein–protein interaction network analysis reveals pathways significantly enriched specifically in the FA-deficient cells, in response to MMC

crosslinking agents compared to the ΔFANCC single mutant, as shown by clonogenic survival after treatment with MMC, cisplatin or diepoxybutane (DEB) (Fig. 3b–d). Interestingly, we did not observe the same effect on survival when we compared WT cells to ΔUSP48 cells, although a slight but not significant difference was observed, further validating the results of our screens and the specificity of this genetic interaction for FA-deficient cells. Re-introduction of exogenous wild-type USP48, but not the catalytically inactive C98S USP48 mutant, partially reduced ICL resistance of ΔUSP48ΔFANCC cells (Supplementary Fig. 2c, d), thus indicating that lack of USP48 catalytic activity is important for the increased survival of ΔUSP48ΔFANCC cells. Further confirming that the synthetic rescue was indeed dependent on USP48, when we subjected USP48 to short-hairpin RNA (shRNA) depletion (Supplementary Fig. 2e, f) or carried out USP48 gene inactivation by CRISPR-Cas9 editing by using a different single guide (sg)RNA targeting a different exon (Supplementary Fig. 2g, h) in ΔFANCC cells, we observed similar results. We also tested the effect of USP48 loss on MMC

sensitivity of ΔFANCG and ΔFANCD2 cells using CRISPR-Cas9 editing to target USP48. The pooled populations of FA mutant cells targeted for USP48 displayed reduced USP48 protein (Supplementary Fig. 2g) and increased survival to MMC (Supplementary Fig. 2h), thus confirming the synthetic viability interaction in additional FA backgrounds.

Since monoubiquitylation of components of the FANCI/FANCD2 complex is an important step of ICL repair[3,4] and considering that USP48 is a DUB, we tested whether USP48 depletion could rescue the FANCI/FANCD2 monoubiquitylation defect of ΔFANCC cells. Western immunoblot assays revealed that lack of MMC-induced FANCI/FANCD2 monoubiquitylation in ΔFANCC cells was not restored upon shRNA-mediated depletion of USP48 (Supplementary Fig. 2i), suggesting that USP48 may affect the repair of ICLs downstream of FANCI/FANCD2. Moreover, USP48 depletion in ΔFANCC cells did not correct the defect that these cells had in recruiting FANCD2 to MMC-induced DNA-damage foci, thereby implying that the mechanism of rescue was independent of FANCD2 (Supplementary Fig. 2j).

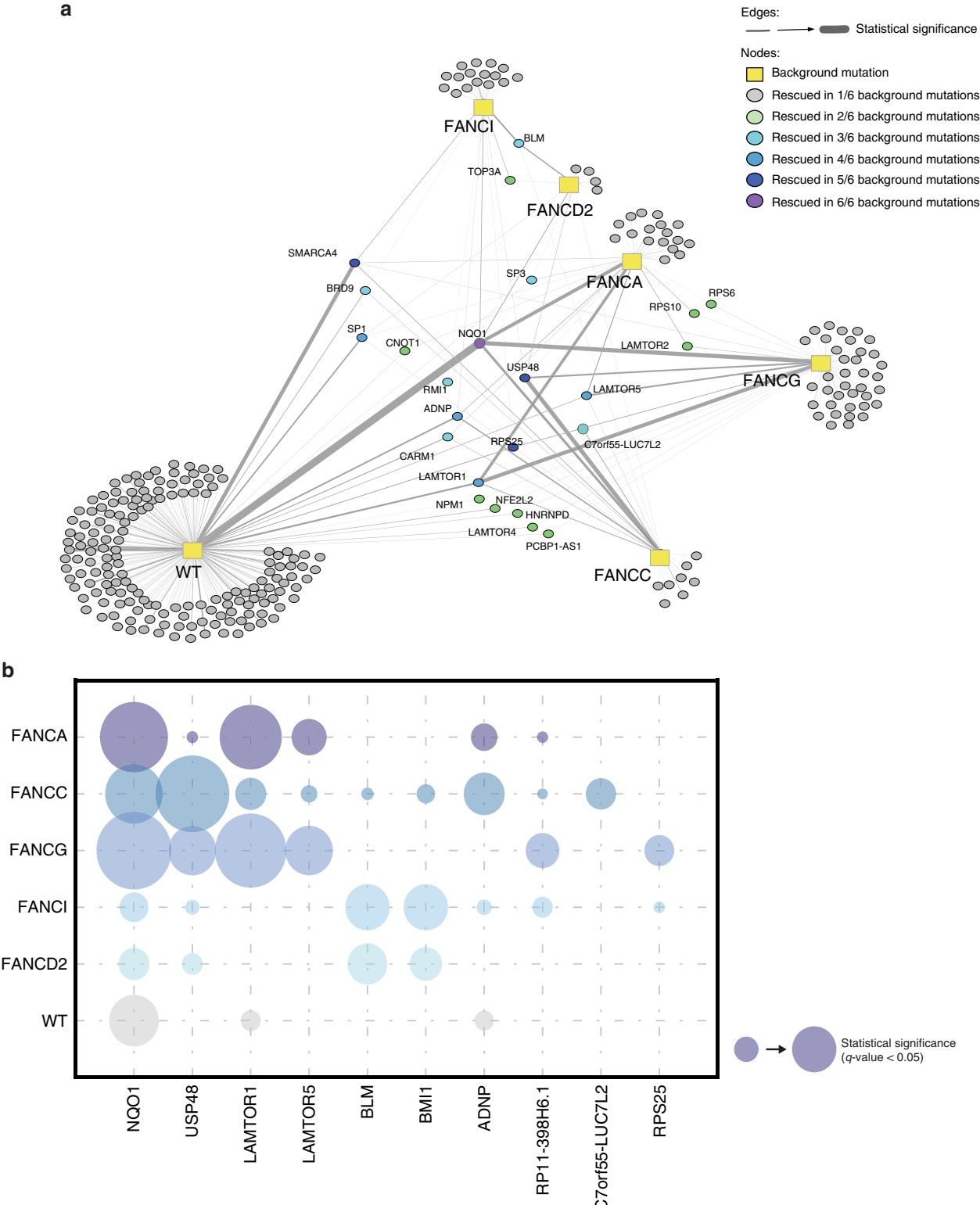

**Fig. 2** Synthetic-viable genes for FA cells. **a** Map of the synthetic viability interaction network identified by insertional mutagenesis screens using wild-type (WT) and five FA-deficient HAP1 cell lines after MMC treatment. Links in this network indicate an increased viability of cell lines with a given mutational background (rectangular nodes) when the respective secondary gene (circles) is also knocked out. A total of 23 genes were found to rescue more than one mutational background (color coded with green to purple), including USP48. **b** Bubble plot showing the frequency and statistical significance of the top ten genes identified in the screens depicted in **a**

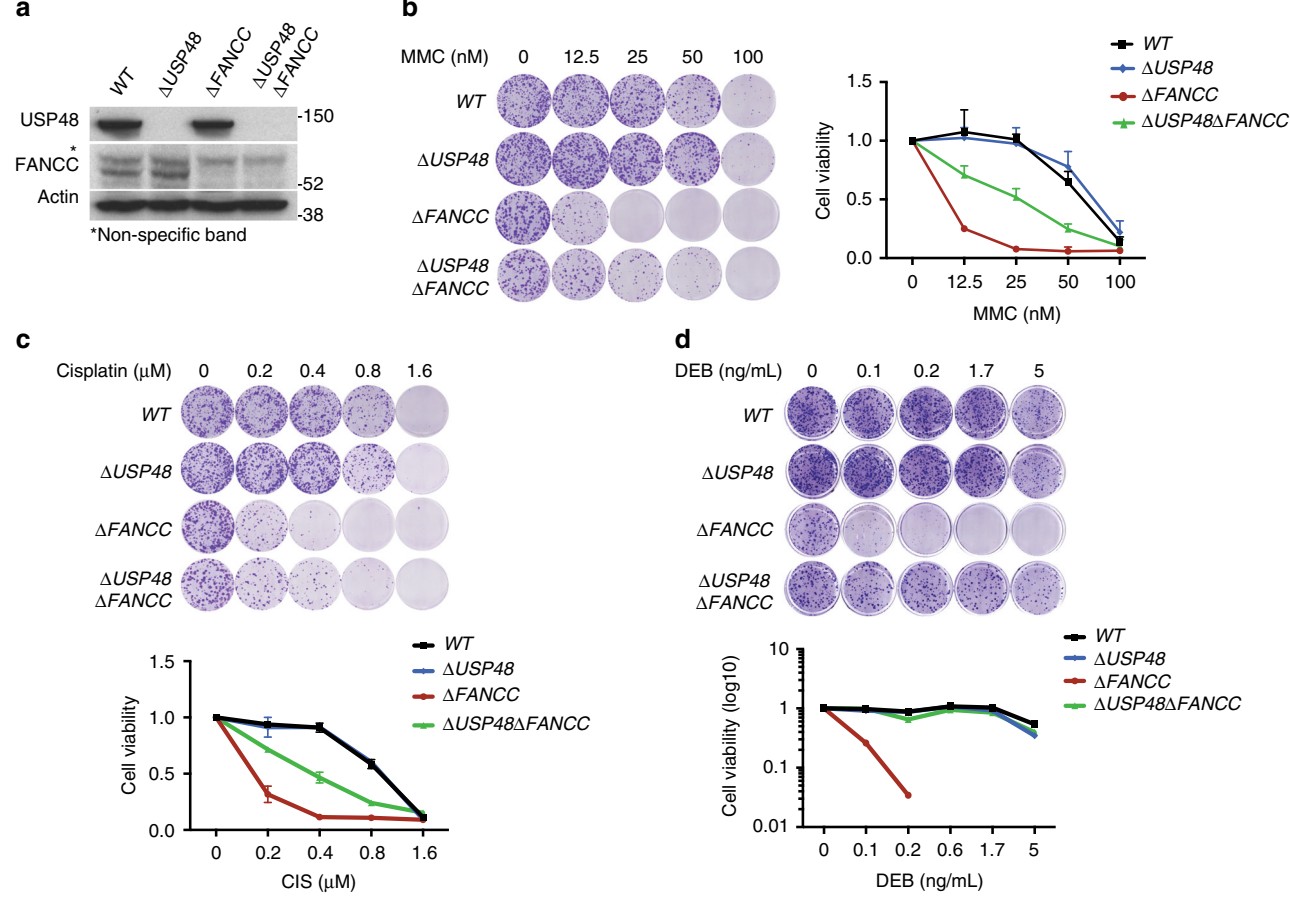

**Fig. 3** USP48 loss partially rescues sensitivity of Δ*FANCC* cells to ICLs. **a** Immunoblot for USP48, FANCC, and actin on the indicated cell lines. Asterisk (*) denotes non-specific band. **b**–**d** Colony formation and subsequent quantification of the indicated cell lines 7 days after treatment with crosslinking agents (Mitomycin C, MMC; Cisplatin; Diepoxybutane, DEB) at the indicated doses. Error bars show S.D. (standard deviation) from two replicates

**USP48 is recruited to sites of DNA damage**. In light of our findings, we next investigated the cellular localization of USP48 and whether it is recruited to sites of DNA damage. GFP-tagged USP48 was expressed in WT and Δ*FANCC* cells with and without MMC treatment. We observed that GFP-USP48, both WT and the catalytically inactive C98S mutant, localized primarily in the nucleus with a pan-nuclear signal (Supplementary Fig. 3). Interestingly, the catalytically inactive C98S mutant, but not WT USP48, formed foci that co-localized with γH2AX foci (arrows in Supplementary Fig. 3). This suggested that USP48 interacts with a substrate at the site of damage and inhibition of its activity entraps the enzyme, hence leading to the formation of USP48 foci. We next tested if USP48 is specifically recruited to sites of DNA damage created by laser micro-irradiation of FANCC-proficient (U2OS) and FANCC-deficient (VU1131) cells (Supplementary Fig. 4a). Indeed, both WT and C98S USP48 accumulated at sites of laser micro-irradiation within 10 min of DNA-damage induction (then declined after a further 10 min), with the mutant protein accumulating more strongly than the WT protein (Supplementary Fig. 4b, c), again suggesting that mutation of the catalytic domain may prevent release of USP48 from its substrate(s) at DNA-damage sites. Furthermore, we found that USP48 DNA-damage accumulation was more efficient in FANCC-deficient cells than in these cells complemented with FANCC, suggesting that USP48 recruitment may be actively suppressed by FANCC and associated FA proteins (Supplementary Fig. 4d, e). Finally, we tested whether USP48 can specifically localize to ICLs by irradiating cells that had been pre-treated with Trioxalen, a psoralen derivative that forms ICLs upon UV

irradiation (Fig. 4a). Notably, we observed efficient recruitment of mutant USP48 in ICL-containing tracks marked by FANCD2 in a substantial fraction of cells, while we were unable to detect WT USP48 recruitment (Fig. 4b). By contrast, we detected DNA-damage accumulation of both WT and mutant USP48 in a substantial fraction of FANCC-deficient cells, with the mutant protein being detectable in laser tracks in a much larger proportion of the cells than the WT protein (Fig. 4b). Collectively, these data indicated that USP48 is recruited to ICLs and that it appears to do so independently of a functional FA-protein assembly.

**USP48 loss promotes repair of MMC-induced damage in FA cells**. We next determined whether the effect of USP48 loss on survival of FA-defective cells after treatment with DNA cross-linking agents correlated with less DNA damage induction and/or increased DNA repair. Hence, we analyzed the formation and clearance of γH2AX foci, a marker for DNA damage lesions, in WT, Δ*USP48*, Δ*FANCC*, and Δ*USP48*Δ*FANCC* cells at different time-points after being treated with MMC for 18 h. Although the number of γH2AX foci were similar in all settings immediately after MMC treatment (0 h after MMC removal), we observed persistence of γH2AX foci at 48 h after MMC removal in Δ*FANCC* cells but not in Δ*USP48*Δ*FANCC* cells (Fig. 4c, d). These findings thus suggested that Δ*USP48*Δ*FANCC* cells repair MMC-induced DNA damage more efficiently than Δ*FANCC* cells. Interestingly, we also observed that Δ*USP48* cells displayed increased clearance of γH2AX foci than WT cells (Fig. 4c, d), suggesting that loss of USP48 improves DNA repair also in ICL-

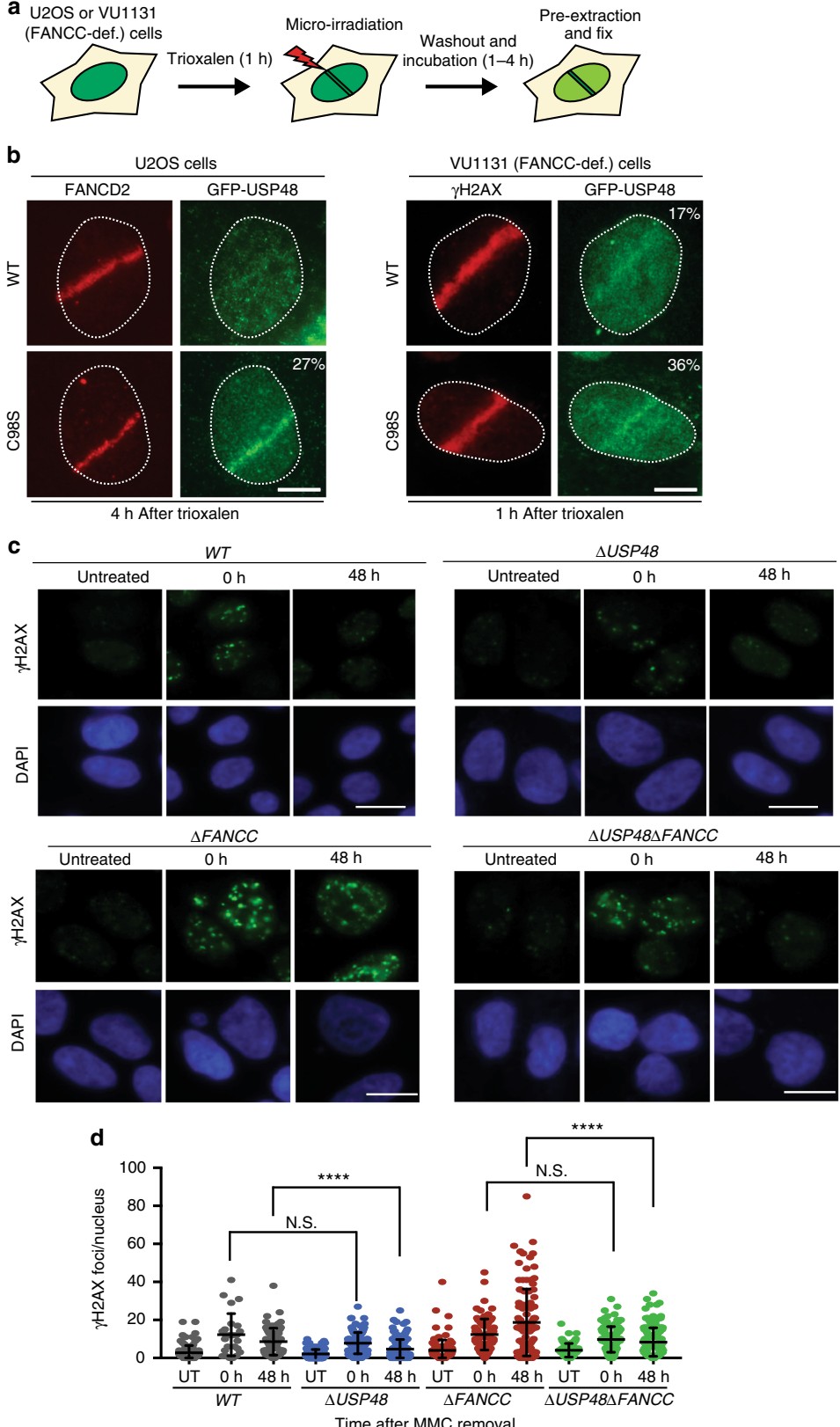

**Fig. 4** USP48 is recruited to ICLs and attenuates clearance of γH2AX foci. **a** Workflow for the recruitment of proteins to ICLs in U2OS and VU1131 cells. **b** Localization of FANCD2 and GFP-USP48 (WT and C98S mutant) to sites of laser micro-irradiation after Trioxalen treatment in U2OS and VU1131 cells. Scale bar = 10 μm. **c** Representative immunofluorescence images after staining for γH2AX on WT, ΔUSP48, ΔFANCC, and ΔUSP48ΔFANCC cells after treatment with MMC (30 nM) for 18 h and then recovery for the indicated time-points. Scale bar = 10 μm. **d** Quantification of γH2AX foci of the indicated cell lines. UT untreated cells. Quantification was performed using the Cell Profiler software. Error bars show mean with S.D. (standard deviation). Statistical significance was determined by Wilcoxon test with $p < 0.05$ as threshold. N.S. not significant; **** = $p < 0.0001$

repair proficient cells, although with a less pronounced phenotypic effect than in a ΔFANCC setting.

**USP48 loss enhances BRCA1 and RAD51 localization in FA cells.** FA-protein-dependent homologous recombination (HR) plays a crucial role in the repair of ICL lesions during S-phase[21]. To assess the potential impact of USP48 loss on HR in ΔFANCC cells, we examined the DNA damage-induced recruitment of BRCA1, a HR factor[22]. Analysis of BRCA1 localization into foci at DNA damage sites revealed that ΔUSP48ΔFANCC cells displayed significantly higher numbers of BRCA1 foci at 24 and 48 h after MMC treatment as compared to ΔFANCC cells (Fig. 5a, b). Furthermore, ΔUSP48ΔFANCC cells displayed a higher number of BRCA1 foci in untreated conditions, which could reflect enhanced BRCA1 recruitment following endogenous replication stress[23]. We did not, however, observe significant differences in the formation or persistence of BRCA1 foci between WT and ΔUSP48 cells (Fig. 5a, b). Since we had found that USP48 can affect the formation or maintenance of BRCA1 foci in ΔFANCC cells, we then addressed whether it also co-localizes with BRCA1 at DNA damage sites. To this end, we investigated the co-localization of BRCA1 with GFP-tagged WT or C98S USP48, after MMC treatment of ΔUSP48ΔFANCC cells. In accord with our results on its co-localization with γ-H2AX (Supplementary Fig. 3), mutant USP48 was observed to co-localize with nuclear foci of BRCA1 (Fig. 5c). Interestingly, in many cases co-localization was not exact with adjacent foci being observed, possibly suggesting competitive activities of the two proteins in DNA repair pathway choice.

BRCA1 is required for loading of RAD51, a protein necessary for pairing of homologous regions and strand exchange during HR[22,24]. We thus measured, by immunofluorescence staining, DNA damage recruitment of RAD51. At 18 h after MMC treatment, recruitment of RAD51 into foci was significantly higher in U2OS cells simultaneously depleted for both FANCC and USP48 by siRNA treatment, compared to cells depleted for FANCC alone (Fig. 5d and Supplementary Fig. 5a–c), implying that USP48 loss restores HR efficiency at replication forks encountering ICL damage in FA-deficient cells. By contrast, USP48 depletion did not enhance MMC-induced RAD51 focus formation in control cells (Fig. 5d and Supplementary Fig. 5c), suggesting that USP48 may inhibit HR processes specifically in an FA-deficient background. Altogether, these findings suggested that USP48 counteracts BRCA1 and RAD51 recruitment at sites of ICLs, most notably when the FA pathway is deficient, thus preventing efficient HR.

To explore potential effects of USP48 on HR processes, we used the well established Traffic Light Reporter (TLR) system in human U2OS cells[25,26]. Our results showed that depletion of USP48 reduced HR levels in both WT and FANCC-depleted cell backgrounds when compared to control cells (Supplementary Fig. 5d). While this did not correlate with the increased RAD51 recruitment we observed upon USP48 depletion after induction of DNA ICLs, we note that the direct generation of DNA double-strand breaks by the endonuclease I-SceI in this system renders it quite different from the HR-repair templates generated via ICL processing in S-phase. As an alternative measurement for homology-based repair, we assessed RPA recruitment to chromatin as an indicator of end-resection. We did not observe a major effect of USP48 depletion on RPA chromatinization upon induction of camptothecin (CPT)-induced replication stress, although a marginally significant increase of RPA foci was evident when both USP48 and FANCC were depleted compared to when FANCC was depleted alone (Supplementary Fig. 5e).

**USP48 loss rescues ICL sensitivity of FA cells through BRCA1.** Since ΔUSP48ΔFANCC cells exhibited markers of enhanced ICL-dependent HR (as evidenced by BRCA1 and RAD51 foci) compared to ΔFANCC cells, and because USP48 inactivation alleviated the sensitivity of ΔFANCC cells to DNA crosslinking agents, we determined whether this effect on cell survival depended on BRCA1. Indeed, shRNA-mediated depletion of BRCA1 reversed the resistance of ΔUSP48ΔFANCC cells to MMC or cisplatin, as shown by clonogenic survival assays (Fig. 6a, b, marked by green frame), although residual resistance was still observed when compared to BRCA1-depleted ΔFANCC cells (Fig. 6b, marked by red frame), possibly because of incomplete shRNA-mediated depletion resulting in residual BRCA1 activity, or reflecting an additional BRCA1-independent effect of USP48. These results were thus in line with a model in which USP48 functions, at least in part, to counteract BRCA1 activity.

To explore the potential involvement of BRCA1 ubiquitin ligase activity[27], we reconstituted BRCA1-depleted ΔUSP48ΔFANCC cells with wild-type BRCA1 (WT) or with a mutant form of BRCA1 carrying a point mutation (I26A), previously shown to affect its ubiquitin ligase activity[28,29]. Notably, we found that both WT and I26A BRCA1 were able to alleviate the hypersensitivity of BRCA1-depleted cells towards MMC treatment (Supplementary Fig. 5f, g), suggesting that the E3 ubiquitin ligase activity of BRCA1 might not be crucial for the synthetic rescue phenotype observed upon loss of USP48 in a FANCC-deficient background.

**Loss of USP48 affects histone H2A ubiquitylation.** We next asked what could be the substrate of USP48 in the context of ICL repair. Ubiquitylation of RPA has been recently implicated in the proper recruitment and resolution of RAD51 accumulation at sites of DNA ICL repair, in a manner dependent on the RFWD3 ubiquitin ligase[30,31]. Interestingly, RFWD3-deficient cells are hypersensitive to DNA crosslinking agents, thus supporting a model where USP48 might counteract the effect of RFWD3 on RPA and RAD51 recruitment. However, we did not observe a significant impact of USP48 siRNA depletion on the levels of ubiquitylation of GFP-tagged RPA70 in U2OS cells treated with camptothecin (Supplementary Fig. 5h, i). Based on these findings, we conclude that it is unlikely that USP48 counteracts RNFWD3/RNF4/PRP19-mediated ubiquitylation of RPA70.

USP48 has been reported to be recruited on chromatin and be associated to modified histones[32]. Histone ubiquitylation and especially H2A/H2B ubiquitylation is involved in repair of DNA double-strand breaks[33,34], and various sites have been identified as ubiquitin substrates that are ubiquitylated by different ubiquitin E3 ligases. We thus decided to investigate whether histone ubiquitylation levels are affected by loss of USP48. To this end, we immunoprecipitated Flag-tagged H2A from WT, ΔUSP48, ΔFANCC, and ΔUSP48ΔFANCC cells following MMC treatment and blotted for ubiquitin (Fig. 6c and Supplementary Fig. 5j). Immunoprecipitation of Flag-tagged WT H2A, followed by immunoblotting against protein-conjugated ubiquitin chains using FK2 antibody showed a small but noticeable increase in poly-ubiquitylated forms of H2A in double ΔUSP48ΔFANCC cells compared to ΔFANCC cells, and increase that was further enhanced by MMC treatment (Fig. 6c). Immunoprecipitation of a mutant form of H2A that lacks the ubiquitin target sites K5-9-118-119-125-127-129 (K5-9-118-119-125-127-129R mutant) showed almost no poly-ubiquitylation, implicating at least one of these sites as being affected by USP48 loss (Fig. 6c, residual bands for the immunoprecipitated mutant H2A in the ΔUSP48 ΔFANCC background likely correspond to ubiquitylated H2A at RNF168-dependent sites K13-15, which were not targeted for mutation, thus not excluding their contribution in the observed poly-ubiquitylated H2A profile).

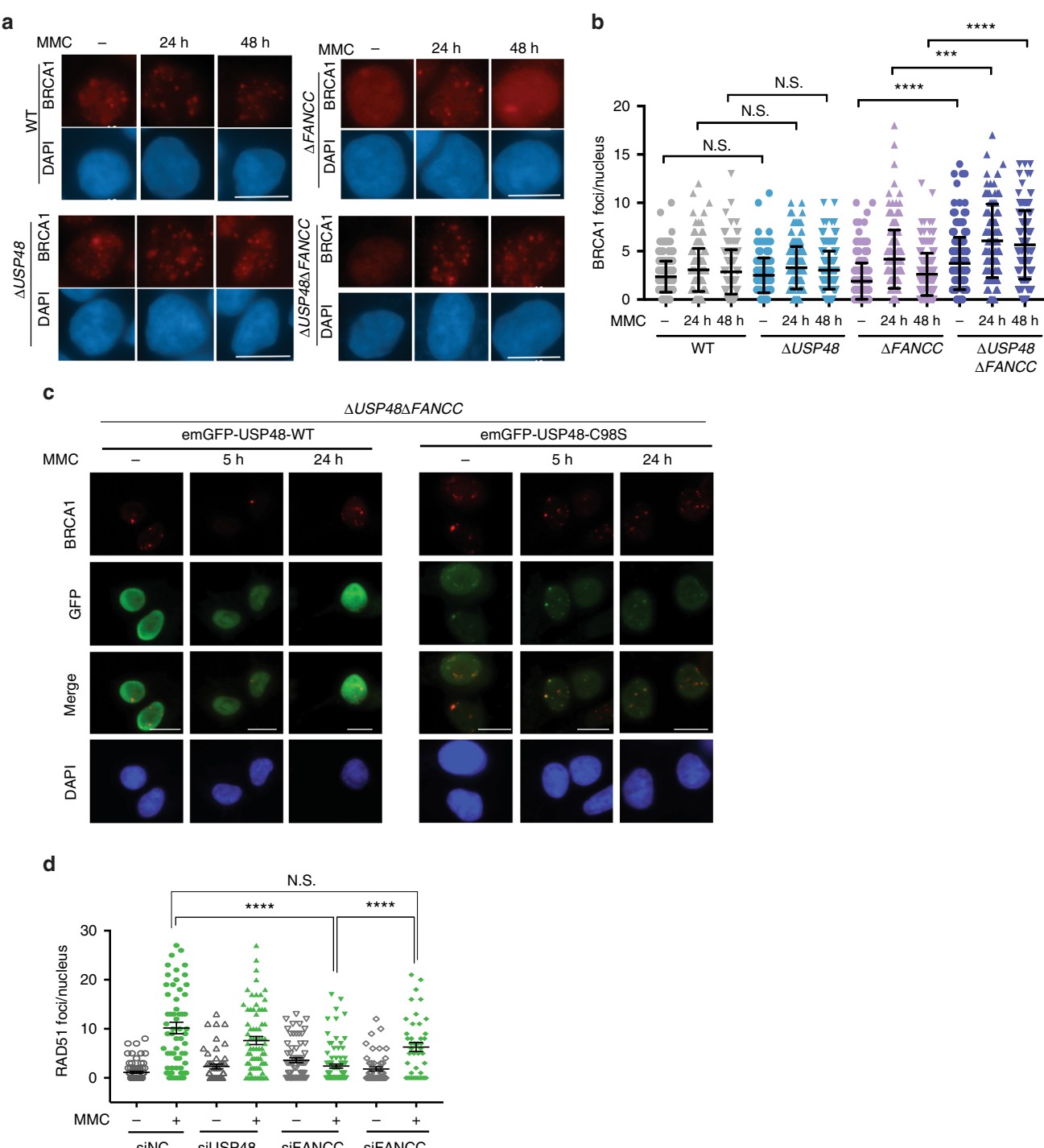

**Fig. 5** FA cells lacking USP48 show enhanced recruitment of HR markers. **a** Representative immunofluorescence images of WT, Δ*USP48*, Δ*FANCC*, and Δ*USP48*Δ*FANCC* cells after staining for BRCA1 following MMC treatment (25 nM) for the indicated times. Scale bar = 10 μm. **b** Quantification of BRCA1 foci in WT, Δ*USP48*, Δ*FANCC*, and Δ*USP48*Δ*FANCC* cells. Error bars show mean with SD (standard deviation). Statistical significance was determined using the Mann–Whitney test with $p < 0.05$ as a threshold. N.S. not significant; *** = $p < 0.001$; **** = $p < 0.0001$. **c** Immunofluorescence images after staining for BRCA1 and GFP on Δ*USP48*Δ*FANCC* cells after treatment with MMC at the indicated time-points. Cells were transiently transfected with emGFP-USP48-WT and emGFP-USP48-C98S. **d** Quantification of RAD51 foci in WT, *siUSP48*, *siFANCC*, and *siUSP48siFANCC* cells treated with MMC (100 nM) for 18 h. Quantification was performed using the ImageJ software. Error bars show mean ± S.E.M. (standard error of the mean). Statistical significance was determined using the Mann–Whitney test with $p < 0.05$ as a threshold. N.S. not significant; **** = $p < 0.0001$

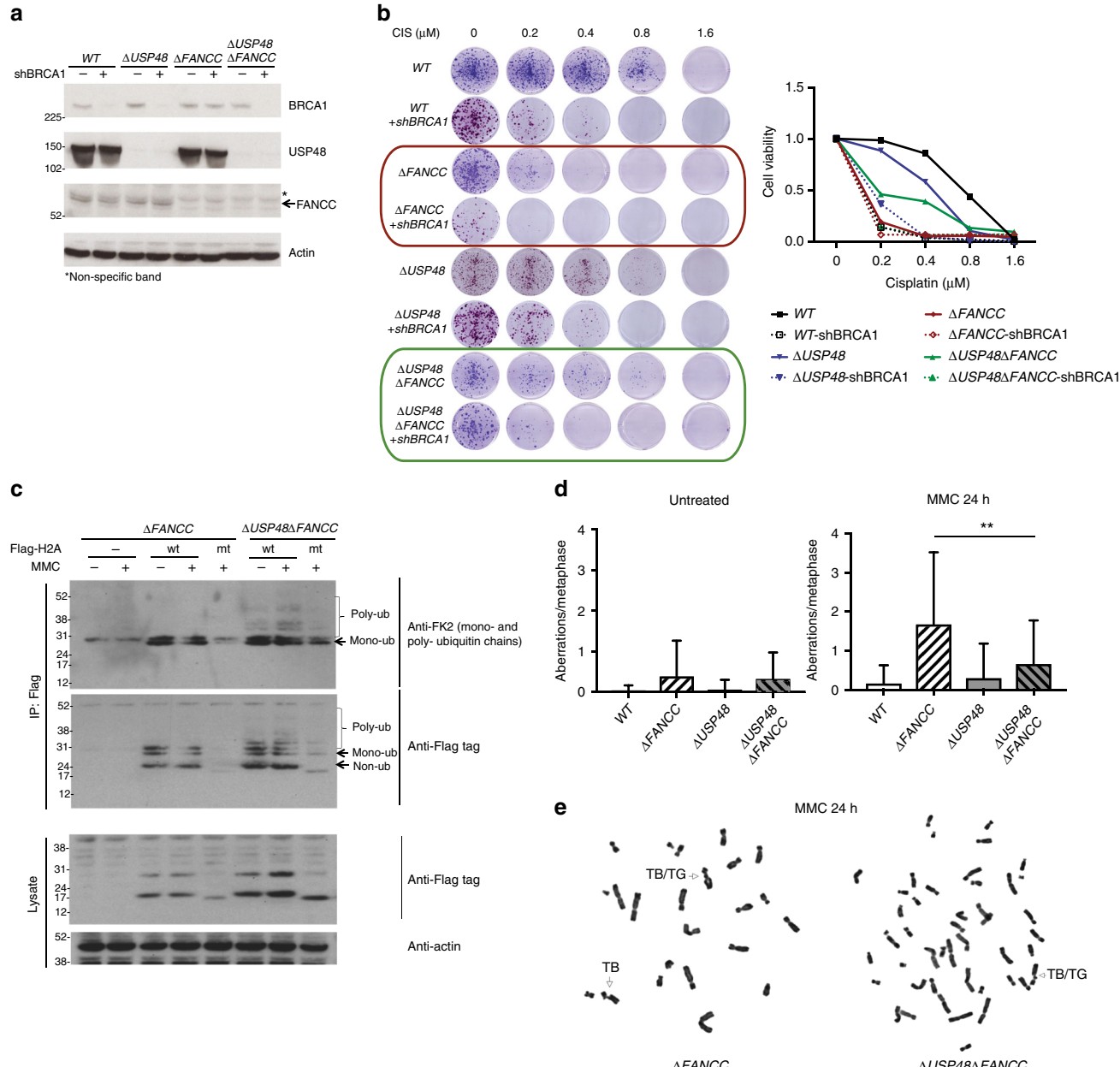

**Fig. 6** USP48 acts on H2A and reduces chromosomal aberrations in FA cells. **a** Immunoblot for BRCA1, USP48, FANCC, and actin on the indicated cell lines. Asterisk (*) denotes non-specific band. **b** Colony formation and quantification of cell survival of the indicated cell lines 7 days after treatment with MMC at the indicated doses. **c** Anti-Flag-H2A immunoprecipitation probed against FK2, Flag and actin for the indicated cell lines. Lower bands correspond to monoubiquitylation modifications and higher bands correspond to poly-ubiquitin chains. Cells were transfected with wt = wild-type and mt = mutant (K5-9-118-119-125-127-129R) H2A and then treated with MMC (50 nM) for 4 h. **d–e** Plot of mean number of chromosomal aberrations (**d**) and representative images from metaphase spreads (**e**) of wild-type (WT), ΔFANCC and ΔUSP48ΔFANCC cells after treatment with MMC (25 nM) for 24 h. TB/TG: chromatid break/chromatid gap. Error bars show SD. Statistical significance was determined using the Mann–Whitney test with $p < 0.05$ as a threshold. ** = $p < 0.01$

These data raise the possibility that USP48 might at least in part affect the ICL sensitivity of FANC-gene deficient via influencing H2A ubiquitylation.

**Genomic instability of FA cells is reduced upon USP48 loss**. A hallmark of FA-defective cells is increased chromosomal aberrations upon treatment with crosslinking agents[3,4], manifested as breaks and radials on mitotic chromosomes. As we had found that USP48 loss enhances HR in ΔFANCC cells, we determined whether it could also alleviate chromosome breaks in ΔFANCC cells. Indeed, analysis of metaphase spreads from cells treated

with MMC showed that chromosomal aberrations were significantly reduced in ΔUSP48ΔFANCC cells compared to ΔFANCC cells, the latter of which exhibited a high number of breaks after MMC treatment compared to WT cells (Fig. 6d; representative images of chromosomal aberrations shown in Fig. 6e). This result thus indicated that absence of USP48 can alleviate the increased genomic instability of FA-defective cells.

## Discussion

In the current study, we have taken an unbiased approach to reveal an extended network of synthetic rescue interactions for

FA deficiency, providing a resource for potential therapeutic targets. Among these, we have selected and validated a synthetic-viable interaction between the deubiquitylating enzyme USP48 and FA genes. We have shown that USP48 loss specifically improves the survival of FA-defective cells to DNA crosslinking agents, and that this is associated with enhanced recruitment of the HR proteins BRCA1 and RAD51 to DNA damage sites and reduced chromosomal instability. This implies that USP48 counteracts BRCA1 activity or indirectly prevents conditions permissive for HR.

Our data indicate an early role for USP48 at sites of DNA damage, as shown by its rapid recruitment at irradiated sites, likely targeting substrates at or flanking sites of DNA lesions. Interestingly, we have found that recruitment and retention of USP48 at ICLs appears to be negatively modulated by its catalytic activity and by the presence of FA proteins, suggesting that FA-protein assembly at DNA damage sites serves to likely counteract USP48 recruitment and activity at sites of DNA ICLs to promote HR and genome integrity. Our findings thus support a role for USP48 in regulating the balance between different repair pathways at the site of damage.

While our manuscript was under review, a new study was published showing that USP48 is recruited to sites of double-strand breaks and that its depletion leads to increased RAD51 recruitment and enhanced BRCA1-mediated DNA end-resection[35]. Although not in the context of interstrand crosslink repair, these results are complementary and similar to what we observed, and further highlight that USP48 plays a role(s) in DNA repair. Notably, the authors in this study showed the activity of USP48 counteracts BRCA1-mediated ubiquitylation of H2A K127/129 and thereby preventing chromatin remodeling and resection during double-strand break repair after IR and camptothecin treatment[35]. Significantly, however, our studies in USP48-depleted FA-gene-deficient cells have indicated that both wild-type and I26A BRCA1 were able to alleviate the hypersensitivity of BRCA1-depleted FA-gene deficient, USP48-depleted cells towards MMC treatment. This suggests that the effect of BRCA1 under this setting is unlikely to operate via effects on reversing BRCA1-mediated ubiquitylation of histone H2A or other proteins. However, since the role of E3 ubiquitin ligase activity of BRCA1 is a subject of debate[28,29,36], more experiments are needed to better understand its involvement or not in the mechanism of the genetic interaction between USP48 and FA genes during crosslink repair. It is important to note that our data do not exclude the contribution of other H2A sites in the mechanism by which USP48 loss counteracts the ICL hypersensitivity of FA cells. Since crosstalk between histone marks has also been reported[37], another possible mechanism is that USP48 regulates ubiquitylation of multiple H2A sites by trimming poly-ubiquitin chains, thus performing a balancing act between different DNA repair pathways in combination with the activity of E3 ubiquitin ligases such as RNF168 or others.

An important step in ICL repair is unhooking of the DNA lesion, which happens with the contribution of various nucleases, including FAN1, MUS81-EME1, XPF-ERCC1, SLX1, and the scaffold protein SLX4, all thought to assemble into a large structure-specific endonuclease complex[21]. Importantly, SLX4 and FAN1 both contain UBZ domains that recognize ubiquitin and play important roles in their recruitment and function that is also dependent, at least partially, on FA proteins, especially monoubiquitylated FANCD2/FANCI[38–41]. Our data demonstrate that loss of USP48 does not restore FANCI/FANCD2 mono-ubiquitylation or FANCD2 recruitment at ICLs, but if USP48 targets one or more sites on H2A that can be recognized by these nucleases, then loss of USP48 might bypass the requirement of

the FA proteins and allow the recruitment of FAN1 or SLX4 and subsequent unhooking of the ICL in an FA-deficient background.

Recently, USP48 has also been shown to promote the stability of Mdm2 that in turn results in enhanced degradation of p53[42], which has been associated to FA cell death[43]. However, HAP1 cells, which have used in various parts of our study, have a p53 mutation, which likely affects p53 normal function, as we have shown in a previous study[18] but also as implied by the fact that p53 is not retrieved as a significant suppressor-hit from any of our screens. Moreover, we have shown similar effects upon USP48 depletion in U2OS cells, which possess wild-type p53. We thus, conclude that the function of USP48 in the context of ICL repair and synthetic rescue in an FA-deficient background is unlikely to be through its effects on p53.

Importantly, the results of the present study show that loss of USP48 improves DNA repair and prevents genomic instability of FA-defective cells, thus highlighting the potential of developing USP48 inhibitory molecules as novel therapeutic approaches that could potentially alleviate the phenotypes of FA patients. In this regard, it will be of interest to employ existing FA mouse models to investigate whether loss or inhibition of USP48 could alleviate disease phenotypes.

## Methods

**Cell lines and culture conditions**. Human HAP1 cells were obtained from Horizon Discovery, they were grown in Iscove's Modified Dulbecco's Medium (IMDM) from GIBCO®, containing L-Glutamine and 25 mM HEPES and supplemented with 10% fetal bovine serum (FBS) and 1% Penicillin/Streptomycin (P/S). HEK293T cells were obtained from the CRUK Cell Facility, they were used for virus production, expanded in Dulbecco's modified Eagle medium (DMEM) and supplemented with 10% FBS. U2OS cells were originally obtained from ATCC cell repository, they were cultured in DMEM (Sigma-Aldrich), supplemented with 10% FBS. Human mCherry-Geminin-expressing U2OS[44], FANCC-deficient VU1131 and complemented VU1131 cells (a gift from Josephine Dorsman, VU Medical Center Amsterdam) were cultured in DMEM, supplemented with antibiotics and 10% fetal calf serum. VU1131 cells were grown with G418 (300 µg/mL). All cells were grown at 37 °C in a 3% oxygen and 5% CO$_2$ atmosphere. All cell lines used in this publication were tested negative for mycoplasma contamination using the MycoAlert™ Mycoplasma Detection Kit. They were all authenticated by the specified providers and, furthermore, they are not listed as commonly misidentified by ICLAC.

**CRISPR-Cas9-mediated gene editing**. ΔUSP48 HAP1 cells were purchased from Horizon Genomics (clone HZGHC000915c012). CRISPR-Cas9 knockouts of FANCA, FANCC, FANCG, and FANCI were generated in collaboration with Horizon Genomics. Sequences for gRNAs were designed by Horizon Genomics or with the use of http://crispr.mit.edu/ and https://www.deskgen.com/landing/, respectively. Sequences of gRNAs used were:

FANCA: 5'-CGGGATGGTTGCCTCTAGCG-3';
FANCC: 5'-GCCAACAGTTGACCAATTGT-3';
FANCG: 5'- GAATGACCGGCTCGTTCGAC-3';
FANCI: 5'-GTATCCAGTTGGTGGAATCG-3';
USP48ex1: 5'- TCGAGACCGCTTACCGGCATC-3'
USP48ex6: 5'-GCTTAGACTCTCTGCCACAC-3'

**Sanger sequencing**. Genomic DNA was extracted using the Qiagen Biotech DirectPCR Lysis Reagent (Cell) according to the manufacture's protocol. Genomic regions around the gRNA-targeted sequences were amplified using the following primer pairs:

FANCA-For: 5'-ATACTGAGCAAACTCTAACAGGGAA -3';
FANCA-Rev: 5'- GGCATTTTAAACAGCAAGTCTTTGG-3';
FANCC-For: 5'-CAAACCTACACACACATACATGGAC-3';
FANCC-Rev: 5'-ACTAAACAAGAAGCATTCACGTTCC-3';
FANCG-For: 5'- GTTGTCACTGAGGATCAATCCTTTT-3';
FANCG-Rev: 5'- TTCACCCTTCCTAAGTCCGCTT-3';
FANCI-For: 5'-CTTTTTCAAAGCCCTTAACCATTGC-3';
FANCI-Rev: 5'-CCCTCAACAAATTACAAACCCTCAA-3';
USP48-For: 5'- GATGGGAACCCAAACCTTCCTAAAG-3';
USP48-Rev: 5'- CTCGGGAGGCGTTCCTGG-3'.

The following sequencing primers were used:
FANCA: 5'-GGCATTTTAAACAGCAAGTCTTTGG-3';
FANCC: 5'-ACTAAACAAGAAGCATTCACGTTCC-3';
FANCG: 5'- GTTGTCACTGAGGATCAATCCTTTT-3';
FANCI: 5'-CTTTTTCAAAGCCCTTAACCATTGC-3';

USP48: 5'- GATGGGAACCCAAACCTTCCTAAAG-3'.

PCR amplification conditions were as follows: heat lid 110 °C; 94 °C 2 min; loop 35 × (94 °C 30 s; 55 °C 30 s; 68 °C 1 min) 68 °C 7 min. Frameshift mutations were identified using Nucleotide BLAST against the reference genome GCF_000001405.33.

**Plasmids.** GFP-WT and GFP-C98S mutant USP48 constructs were generated via Gateway cloning. WT and C98S mutant USP48 cDNA sequences were amplified by PCR using primers complementary to attB sites and partially to cDNA sequence and cloned into pDONR221 vector. Following reaction with L recombinase (Invitrogen, Thermo Fisher Sci) WT and C98S mutant USP48 cDNA sequences were cloned into the pcDNA6.2/N-EmGFP-DEST destination vector (Invitrogen, Thermo Fisher Sci). Correct sequence and in-frame N-terminal expression of the GFP tag was confirmed by Sanger sequencing. A mammalian expression plasmid encoding 3 × HA-Ubiquitin was previously generated together with the laboratories of Yossi Shiloh (Sackler School of Medicine, University of Tel Aviv) and Moshe Oren (Weizmann Institute of Science). pCDNA3.1-Flag-H2A (Addgene plasmid # 63560) and pCDNA3.1-Flag-H2A K5-9-118-119-125-127-129R (Addgene plasmid # 63565) were a gift from Titia Sixma[34]. pEGFP1-C1-BRCA1 and pCIN4-Flag-BARD1 plasmids were a kind gift from Jiri Lukas (Novo Nordisk Foundation Center for Protein Research, University of Copenhagen). To generate the I26A mutant construct, the Q5® site-directed mutagenesis kit (NEB) was used according to the manufacturer's instructions. The mutagenesis primers were designed using the NEBaseChanger™ Tool (http://nebasechanger.neb.com/). The following primer sequences were used:

BRCA1 I26A For: 5'-AGAGTGTCCCgcCTGTCTGGAG-3',
BRCA1 I26A Rev: 5'-AAGATTTTCTGCATAGCATTAATG-3'

**Plasmid transfection.** Transfection of pcDNA6.2/N-emGFP USP48 WT and C98S vectors was carried out using Xfect transfection reagent (Takara, Clontech) according to manufacturer's protocol. Briefly, HAP1 cells at 60% confluency in six-wells plates were transfected with 8 µg of plasmid in antibiotics-free medium containing 10% FBS. The next day, cells were trypsinized and seeded for colony formation or immunofluorescence assays.

**BRCA1 siRNA transfection.** Transfection of HAP1 cells with siBRCA1 and non-targeting control siRNA was done using Dharmafect-1 (Dharmacon) reagent according to manufacturer's instructions. For expression of BRCA1 constructs, Effectene transfection reagent (Qiagen) was used the next day following siRNA transfection according to manufacturer's protocol.

siRNA sequences targeting 3'UTR of BRCA1 were synthesized by Dharmacon:
siBRCA1 3'UTR-1: GCUCCUCUCACUCUUCAGU
siBRCA1 3'UTR-2: AAGCUCCUCUCACUCUUCAGU

**BRCA1 knock-down by shRNA.** HAP1 cells were infected with the virus-containing supernatant in the presence of polybrene (final concentration 8 µg/mL), diluted 1:3. Infected cells were selected using puromycin (2 µg/mL; Sigma-Aldrich) for 48 h.

**DsiRNAs.** Pre-designed dicer-substrate short interfering 27-mer RNAs (DsiRNAs) targeting USP48 (design ID hs.Ri.USP48.13.1 and hs.Ri.USP48.13.2), FANCC (design ID hs.Ri.FANCC.13.1 and hs.Ri.FANCC.13.2) or RBBP8/CtiP (design ID hs.Ri.RBBP8.13.1) were from Integrated DNA Technologies. Cells were transfected with 5 nmol DsiRNAs using Lipofectamine RNAiMAX (Thermo Fisher Scientific) according to the manufacturer's instructions.

**Quantitative reverse transcription PCR (RT-PCR).** Cells were harvested from which RNA was isolated using Trizol extraction (following the manufacturer's instructions). RNA was treated with 1 µL DNase (Sigma) and then reverse transcribed with the SuperScript III Reverse Transcriptase protocol (Invitrogen) to obtain cDNA. An amount of 1 µg of cDNA template was used for the qRT-PCR using SYBR Green qPCR Mastermix (Qiagen). Analysis was performed out in triplicates using expression of GAPDH for normalization of data. The PCR was performed on a 7900HT Fast Real-Time PCR System (Applied Biosystems). The following primers were used:

BRCA1: 5'-TCAACTCCAGACAGATGGGAC-3'; 5'-GGCTGTGGGGTTTCTCAGAT-3', GAPDH: 5'-CGAGCCACATCGCTCAGACA-3'; 5'-GGCGCCCAATACGACCAAAT-3'.

**Dose–response curves.** Dose–response curve for mitomycin C (MMC), was performed in 96-well plates by seeding 1000 cells per well, the day before treatment. The following day, compounds were added at twofold serial dilutions. Four days after treatment with compounds, cell viability was measured using CellTiter-Glo (Promega).

**Colony formation assays.** Cells were seeded in six-well plates the day before exposure to compounds (1000 cells/well). The next day compounds were added at the indicated concentrations. Three days after, compound-containing medium was changed with fresh compound-free medium. Cells were left in culture until visible colonies appeared (7–10 days). Colonies were then fixed in 3.7% formaldehyde in phosphate-buffered saline (PBS) for 1 h, washed in PBS and stained with 0.1% crystal violet solution in PBS supplemented with 10% ethanol for 1 h, followed by washing twice with $H_2O$. For quantification, crystal violet was extracted using 50% EtOH, and absorbance was measured at 595 nm.

**Laser micro-irradiation.** U2OS cells were grown on 18 mm coverslips and sensitized with 6 µM Trioxalen for 1 h, or 10 µM 5′-bromo-2-deoxyuridine (BrdU) for 24 h as described[44]. The cells were placed in a Chamlide TC-A live-cell imaging chamber that was mounted on the stage of a Leica DM IRBE widefield microscope stand (Leica, Wetzlar, Germany) integrated with a pulsed nitrogen laser (Micropoint Ablation Laser System; Photonic Instruments, Inc., Belfast, Ireland). The pulsed nitrogen laser (16 Hz, 364 nm) was directly coupled to the epifluorescence path of the microscope and focused through a Leica 40 × HCX PLAN APO 1.25–0.75 oil-immersion objective. The growth medium was replaced by $CO_2$-independent Leibovitz's L15 medium supplemented with 10% FCS and penicillin–streptomycin and cells were kept at 37 °C. The laser output power was set to 75 to generate strictly localized sub-nuclear DNA damage. Cell were micro-irradiatied (two iterations per pixel) within 35 min using Andor IQ software. Following micro-irradiation, cells were incubated for the indicated time-points at 37 °C in Leibovitz's L15 and subsequently fixed with 4% formaldehyde before immunostaining.

**Immunofluorescence.** BRCA1, γH2AX, and GFP staining: Cells were seeded on coverslips in 24 well plates, at a density of $2 \times 10^4$ cells/mL. For BRCA1 staining, soluble proteins were pre-extracted with 0.2% Triton X-100 for 1 minute at room temperature. Cells were then fixed in 4% PFA and permeabilized with 0.2% Triton X-100. After blocking in 10% FCS in PBS for 1 h at room temperature, cells were incubated with primary antibodies overnight and with secondary AlexaFluor antibodies for 1 h at room temperature. DNA was stained with DAPI.

RAD51 and FANCD2 staining: Cells were washed with PBS containing 0.1% Tween20 (PBST), fixed 20 min with 2% paraformaldehyde (w/v) in PBS and blocked in PBST containing 5% bovine serum albumin (BSA) (w/v). Primary antibody incubation with anti-Rad51 (H-92) antibody (Santa Cruz, sc-8349) diluted 1:100 or anti-FANCD2 antibody (Santa Cruz, sc-20022) diluted 1:100 in 5% BSA PBST was performed at 4 °C overnight. After washing with PBST, cells were incubated with AlexaFluor 488-conjugated secondary antibody (Molecular Probes, A11034) diluted 1:1000 in 5% BSA PBST and counterstained with DAPI (2 µg/mL). After washing in PBST the cells were mounted using Vectashield (Vector Labs).

Labeling of laser micro-irradiated cells: Cells were pre-extracted with 0.25% Triton-X-100 (Serva, Heidelberg, Germany) in cytoskeletal (CSK) buffer (10 mM Hepes-KOH, 300 mM Sucrose, 100 mM NaCl, 3 mM $MgCl_2$, pH 7.4) on ice for 1 or 2 min and subsequently fixed with 2% formaldehyde in PBS for 20 min at room temperature. Cells were blocked for 15 min in PBS with 0.5% BSA and 0.15% glycin (PBS+). Antibody steps and washes were in PBS+. The primary antibodies were incubated overnight at 4 °C. Detection was done using goat anti-mouse or goat anti-rabbit Ig coupled to Alexa 488, 546 or 647 (1:1000; Invitrogen Molecular probes). Samples were incubated with 0.1 µg/mL DAPI and mounted in Polymount.

**Microscopy of immunofluorescence.** γH2AX imaging: Images were taken with a Leica DMI 6000B microscope with an HBO lamp with a 100-W mercury short arc UV-bulb light source and six filter cubes, with only three cubes necessary to produce excitations at wavelengths of 360, 488, and 555 nM. Images were captured at each wavelength sequentially with a Plan apochromat HCX 100 × /1.4 oil objective at a resolution of 1392 × 1040 pixels.

BRCA1 imaging: Images were captured on an AxioImager M2, with three lasers giving excitation at 360, 555 and 488 nM wavelengths. Images at each wavelength were collected sequentially at a resolution of approximately 1024 × 1024 pixels, with a Plan apochromat 100 × /1.4 oil objective.

RAD51 and FANCD2 imaging: Images were captured on a FluoView 1000 confocal microscope (Olympus) through a 100 × UPlanSApo/1.4 Oil objective. Images were collected sequentially at each wavelength (405 nm and 488 nm) at the same resolution as above.

Imaging of laser micro-irradiated cells: Images of fixed samples were acquired on a Zeiss AxioImager M2 or D2 widefield fluorescence microscope equipped with 40 × , 63 × and 100 × PLAN APO (1.4 NA) oil-immersion objectives (Zeiss) and an HXP 120 metal-halide lamp used for excitation. Fluorescent probes were detected using the following filters: DAPI (excitation filter: 350/50 nm, dichroic mirror: 400 nm, emission filter: 460/50 nm), GFP/Alexa 488 (excitation filter: 470/40 nm, dichroic mirror: 495 nm, emission filter: 525/50 nm), mCherry (excitation filter: 560/40 nm, dichroic mirror: 585 nm, emission filter: 630/75 nm), Alexa 555 (excitation filter: 545/25 nm, dichroic mirror: 565 nm, emission filter: 605/70 nm), Alexa 647 (excitation filter: 640/30 nm, dichroic mirror: 660 nm, emission filter: 690/50 nm). Images were recorded using ZEN 2012 software.

**Quantification and analysis of immunofluorescence**. BRCA1 and γH2AX analysis: Quantification of foci (number and intensity) was performed using Cell Profiler software.

RAD51 and FANCD2 foci analysis: Quantification of Rad51 foci was performed in ImageJ using a macro developed by Dr. Richard Butler at the Gurdon Institute Imaging Facility (University of Cambridge), which is essentially a version of the FindFoci plugin specifically optimized for Rad51 and FANCD2 foci quantification and the immunofluorescence conditions detailed above.

Local irradiated areas: Images recorded after micro-irradiation of cells were analyzed using ImageJ software. The average pixel intensity of laser tracks was measured within the locally irradiated area ($I_{damage}$), in the nucleoplasm outside the locally irradiated area ($I_{nucleoplasm}$) and in a region not containing cells in the same field of view ($I_{background}$). The relative level of accumulation expressed relative to the protein level in the nucleoplasm was calculated as follows: $((I_{damage} - I_{background})/(I_{nucleoplasm} - I_{background}) - 1)$ for GFP signals and $((I_{damage} - I_{background}) - (I_{nucleoplasm} - I_{background}) - 1)$ for γH2AX signals. The average reflects the quantification of between 50–150 cells from 2–4 independent experiments.

**Immunoblotting and antibodies**. Cell extracts were prepared in RIPA lysis buffer (NEB) supplemented with protease inhibitors (Sigma) and phosphatase inhibitors (Sigma, NEB). Immunoblots were performed using standard procedures. Protein samples were separated by sodium dodecyl sulfate–polyacrylamide gel electrophoresis (SDS–PAGE) (4–12% gradient gels; Invitrogen) and subsequently transferred onto nitrocellulose membranes. Details of primary antibodies used for western blotting (WB) and immunofluorescence (IF) are described in Supplementary Table 1. Secondary antibodies were used at 1:5000 (HRP-conjugated goat anti-mouse, rabbit or goat IgG from Jackson Immunochemicals) for WB and 1:600 or 1:1000 for IF (AlexaFluorophores).

**H2A immunoprecipitation**. HAP1 cells were transfected in 15 cm dishes with FLAG-H2A constructs using X-fect transfection reagent (Clontech) and 40 h later treated with mitomycin C for 4 h. Cell extracts were prepared in nuclear/chromatin extraction HEPES buffer (HEPES 20 mM pH 7.9, NaCl 420 mM, glycerol 25%, EGTA 1 mM, EDTA 1 mM, DTT 0.1 mM) supplemented with NaF 50 mM, $Na_3VO_4$ 1 mM, PMSF 1 mM, NEM 20 mM, protease inhibitor cocktail (Sigma-Aldrich) and Benzonase endonuclease 25 U/mL (Merck-Millipore). After lysis, extracts were sonicated and protein was quantified by Bradford. One milligram of lysate was used for immunoprecipitation after dialysis with HEPES buffer to a final salt concentration of 150 mM NaCl and then incubated with FLAG M2 Magnetic Beads (Sigma-Aldrich) overnight at 4 °C. Beads were washed in wash buffer (HEPES 50 mM, NaCl 150 mM, EDTA 5 mM, NP-40 substitute 0.2% three times and bound proteins were eluted in 2 × Laemmli Sample Buffer for 20 min with gentle agitation. Eluates were loaded on SDS–PAGE gels for western blot.

**RPA70 immunoprecipitation**. To test whether RPA70 is de-ubiquitylated in an USP48-dependent manner, we co-expressed HA-ubiquitin in either GFP-expressing or GFP-RPA70-expressing U2OS cells (kind gift from John Rouse at University of Dundee), transfected with pre-design DsiRNAs targeting USP48 (from Integrated DNA Technologies) using Lipofectamine RNAiMAX according to the manufacturers' instructions. Seventy-two hours post-tranfection cells were treated or mock-treated with camptothecin (TOP1 inhibitor) 1 μM for 1 h. Next, we prepared cell extracts and assessed GFP immunoprecipitates for HA-ubiquitin and FK2 staining by immunoblotting as described in Schmidt et al.[26] Briefly, U2OS cells expressing GFP or RPA70-GFP were transfected in 10 cm dishes with negative-control or USP48-targeting DsiRNAs using Lipofectamine RNAiMAX (Thermo Fisher Scientific) the night before. The following day cells were transfected with a mammalian expression plasmid encoding 3 × HA-Ubiquitin using TransIT-LT1 transfection reagent (Mirus Bio LLC), and 48 h later treated with camptothecin plus 10 μM MG132 proteasome inhibitor for 1 h. Cell extracts were prepared by scraping cells from PBS-washed plates into ice-cold lysis buffer (20 mM Tris-HCl pH 7.5, 40 mM NaCl, 2 mM MgCl2, 10% glycerol, and 0.5% NP-40) containing EDTA-free protease inhibitor cocktail (Roche) and benzonase (10 μl/mL lysis buffer, Novagen 70664-3). After increasing the salt concentration to ~250 mM NaCl, extracts were rotated for 10–15 min at room temperature, subsequently diluted 1:2 with lysis buffer (without benzonase) and the extracts cleared by centrifugation at 16,000 × g for 45 min at 4 °C. Immunoprecipitation reactions were rotated overnight at 4 °C using GFP-Trap-A beads (ChromoTek, 10 μl per mg protein) followed by five washes with immunoprecipitation buffer (lysis buffer without benzonase containing a final concentration of ~250 mM NaCl). Proteins were eluted from the GFP-Trap-A beads in a 5-10-min incubation step at 95 °C in 1.5 × SDS sample buffer and loaded on SDS–PAGE gels for western blot.

**TLR assay**. . A minimum of 10,000 doubly-transfected (IFP+ and BFP+) cells were scored for each condition in four independent experiments, three of which performed in duplicate using either of two different DsiRNAs targeting USP48 or FANCC[26]. Analyses were conducted using FlowJo (TreeStar). For each DsiRNA treatment, results were normalized (including the negative-control DsiRNA) to cells treated with RNAiMAX only, resulting in relative homologous recombination (HR) and mutagenic end-joining (mutEJ) percentages for each DsiRNA treatment.

Finally, the HR values for each condition were normalized to the total amount of cells in S and G2 phases obtained by flow-cytometry analysis of a subset of DAPI-stained cells transfected only with DsiRNAs and grown in parallel at the same cell density.

**RPA70-GFP chromatinization**. Cells were mock-treated or treated for 1 h with 1 μM camptothecin to induce replication fork collapse, and then processed as described in detail in Forment and Jackson[45]. RPA70-GFP + cells were co-stained with DAPI and analyzed by flow-cytometry using a BD LSRFortessa™ cell analyzer (BD Biosciences).

**Metaphase spreads**. Cells were seeded in 10 cm dishes and treated with MMC for the indicated times. Colcemid (KaryoMAX™, Gibco, Thermo Fisher Scientific) was added at a final concentration of 500 ng/mL 3 h before harvesting. Cells were trypsinized and incubated in KCl at 0.075 M (KaryoMAX™, Gibco, Thermo Fisher Scientific) for 6 min. After centrifugation, cells were resuspended in fixation solution (methanol:acetic acid at 3:1) and incubated for 15 min at room temperature. Centrifugation and re-suspension in fresh fixation solution was repeated two times. Metaphase spreads, slide preparation and measurement of chromosomal aberrations was performed at Karyologic Inc (North Carolina, USA). All analysis was performed in a blinded manner. Fifty metaphases were scored for each condition, providing a total count of the overall chromosome counts, and the total of the different types of aberrations (chromatid gaps, chromosome gaps, chromatid breaks, chromosome breaks, double minutes, tri-radial, quadri-radial, complex re-arrangement, pulverized, and polyploid).

**Genome-wide insertional mutagenesis**. Gene-trap virus was produced in HEK293T using Lipofectamine 2000 Transfection reagent (Invitrogen, Thermo Fisher Scientific)[11]. ΔFANCA, ΔFANCC, ΔFANCG, ΔFANCI, and ΔFANCD2 HAP1 cells were transduced with concentrated retrovirus containing the gene-trap cassette[11]. After integration of the GFP-expressing gene-trap cassette, cells were analyzed by flow-cytometry to measure efficiency of infection and populations with more than 70% GFP-expressing cells were used for treatment with MMC. The control non-selected WT-HAP1 population was generated in Blomen et al.[14]. One-hundred million cells from the mutagenized pools were seeded in 15 cm dishes at a density of 6 million cells per dish. The following day, MMC was added at a concentration that selectively killed FA-deficient cells, leaving only around 5–10% of cells surviving (46 nM for ΔFANCA, ΔFANCC, and ΔFANCG, 40 nM for ΔFANCI and ΔFANCD2). Cells were left to grow for 10 days after which cells were trypsinized and frozen at -80 °C. For preparation of the gene-trapped DNA libraries, genomic DNA was extracted from 30 million cells using the QIAamp DNA mini kit (Qiagen), subjected to digestion with MseI (NEB) and NlaIII enzymes (NEB) and subsequently ligated by T4 DNA ligase (NEB). Digested and ligated fragments were used as templates for inverse PCR with primers targeting the LTR regions of the gene-trap cassette. After amplification and purification of the fragments, the DNA sample was prepared for next generation sequencing (Illumina HiSeq 2000,50 base pair single-read) by the CeMM Biomedical Sequencing Facility (BSF). For the control non-selected WT-HAP1 sample, samples from Blomen et al.[14] were used.

**Statistical analysis**. Gene-trap screen: Bioinformatics analysis of the next generation sequencing data was done in R as described[18]. Insertions were considered mutagenic or disruptive to the gene if they occurred within exons irrespective of their orientation to the corresponding gene or if they were located within introns in sense orientation. Insertions in antisense direction in respect to the gene orientation were considered silent. All mutagenic insertions were summarized independently for each gene. For each gene a one-sided Fisher's exact-test was applied to estimate a significant enrichment of insertions over an unselected control data set.

Immunofluorescence analysis: Statistical analysis of the immunofluorescence data was performed using Prism software. We performed Student $T$-test analysis to determine statistical significances between independent samples, this test requires the assumption of a normal distribution of the groups. When it is not possible to assume a normal distribution, we have performed the Mann–Whitney test.

Metaphase spreads analysis: Statistical analysis was performed using Prism software. We performed a Mann–Whitney test. The significance was indicated as: **** = $p < 0.0001$, *** = $p < 0.001$, ** = $p < 0.01$, * = $p < 0.05$, N.S. = not significant.

**Network analysis**. Background network construction: Network analysis methods were employed to explore the functional context of the genes identified in the FA screens (individual, all combined, and those unique to the screens not found in WT). The background interactome used was that compiled by Menche et al.[15] and contains physical protein–protein interactions that were manually curated from seven different databases, including regulatory interactions (TRANSFAC), Yeast two-hybrid screens (IntAct, MINT), literature curated interactions (IntAct, MINT, BIOGRID, HPRD), metabolic enzyme-coupled interactions (KEGG, BIGG), protein complexes (CORUM), kinase network (PhosphositePlus), and signaling interactions[46]. A gene list from the FA screens (individual, all combined and those unique to the screens not found in WT) was used to seed the network.

Network propagation: To identify the local interactome neighborhoods of the genes identified in the FA screen, we implemented the network propagation method previously published[15]. Briefly, the method simulates how heat would diffuse, with loss, through the network by traversing the edges, starting from an initially hot set of "seed" nodes. At each step, one unit of heat is added to the seed nodes and is then spread to its neighbors. A constant fraction of heat is then removed from each node, so that total heat is conserved in the system. The amount of heat per node eventually converges to a stable value, which can then be used as a measure of its integrated distance to the set of seed nodes. For example, a node connected to several seed nodes would have a higher final heat value than a node that is located far from the seeds.

Identify network neighborhood of seed genes and clustering: The network propagation method described above finds genes, which are close to the set of seed genes in network space. We sort the genes in the network by their network propagation score, and extract genes in the top $N$ ($N = 1000$ here) of the sorted list to define the network neighborhood. This results in a subgraph, which we cluster into groups of highly interconnected genes, using a modularity maximization clustering algorithm[17]. Functional enrichment analysis of the genes in each cluster was performed using the ToppFun function of ToppGene[47] and the ORA function as published[48].

**Data availability**. All data generated or analyzed during this study is included in this published article and its Supplementary Information or from the authors upon reasonable request.

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

## Acknowledgements

The ΔFANCD2 HAP1 cells were kindly provided by Prof Ketan J. Patel (LMB, Cambridge, UK). The shRNA constructs for BRCA1 were a kind gift from Prof Sebastian

Nijman (Ludwig Cancer Research, Oxford, UK). The EGFP-BRCA1 construct was a kind gift from Prof. Jiri Lukas (Novo Nordisk Foundation Center for Protein Research, University of Copenhagen). Dr. Josephine Dorsman (VU Medical Center, Amsterdam) kindly provided VU1131 and complemented VU1131 cells. We thank Dr. Thijn Brummelkamp (NKI, The Netherlands) for providing protocols and the WT-HAP1 insertional mutagenesis data. Ms. Fiorella Schischlik and Dr. Michael Schuster assisted with the analysis of the NGS data. All NGS was performed by the Biomedical Sequencing Facility, CeMM, Austria. G.V. was a recipient of an EMBO Fellowship (ALTF 730-2014) and is supported by an FWF grant to J.I.L. (Project number: FWF29555). M.O. is supported by an FWF grant to J.I.L. (Project number: 29763). L.R.-G. is funded by a Boehringer Ingelheim Fonds Ph.D fellowship. The Loizou lab was funded by a Marie Curie CIG to J.I.L. (Project number: 321602-NonCanATM). CeMM is funded by the Austrian Academy of Sciences. K.M.F. and S.B.R. were partially supported by the National Institutes of Health, Grant UL1TR001442 of CTSA. The lab of H.v.A. is funded by an ERC Consolidator grant. Research in the S.P.J. laboratory is funded by Cancer Research UK (program grant C6/A18796) and a Wellcome Trust Investigator Award 206388/Z/17/Z). Institute core infrastructure funding is provided by Cancer Research UK (C6946/A24843) and the Wellcome Trust (WT203144).

## Author contributions

G.V. designed the experiments with help from L.R.G. G.V. and L.R.G. performed experiments, interpreted results, and prepared figures. G.V. wrote the manuscript with help from L.R.G. F.M.-M., and W.W.W. M.O. and M.M. performed experiments, analyzed data, and interpreted results. J.F.da.S. analyzed data. M.W. performed experiments. J.Mo. provided reagents. S.B.R., K.M.F., and J.M. analyzed data and contributed to writing the manuscript. H.v.A. and S.P.J. interpreted results and contributed to writing the manuscript. J.I.L. designed experiments, interpreted results, and wrote the manuscript.

## Additional information

**Competing interests:** The authors declare no competing interests.

