## [Peer Review File · Nature Communications]

Reviewers' comments:

Reviewer #1 (Remarks to the Author):

"Map of synthetic rescue interactions for the Fanconi anemia DNA repair pathway identifies USP48" by Joanna Loizou and colleagues

This is a short paper describing the identification of genetic interactors with knock-out mutants of human Fanconi Anemia genes. Most intriguingly, the paper describes a genetic suppressor of FA mutant phenotypes – the DUB USP48, as well as providing some details on the mechanism of suppression. This is a tantalizing paper. One immediately thinks of possible next steps – eg what is the USP48 target responsible for suppression or what is the effect of pharmacological inhibition of USP48. Nonetheless, the present finding is in my opinion sufficient in principle for publication in Nature Communications, if the authors address the following points.

Major points:

(1) "Recently the concept of synthetic viability (also termed synthetic rescue or genetic suppression) has emerged as a powerful potential approach to ameliorate defects associated with human genetic diseases⁵⁻⁸." I found this sentence misleading, the concept of genetic suppression is a classical concept in genetics and I would urge the authors to introduce this properly, ie they are using a very classical concept, but putting it to new use.

(2) Fig. 4d is lacking Δ USP48. This would be necessary to judge the suppressive effect in the double ko line. Also, I would find it much more useful if the different cell lines would be plotted in the same graph so that a comparison would be easier.

(3) Fig. 5a-b lack the controls for the shBRCA1 in WT background, as well as in the single mutant backgrounds. This is very important, because if the shBRCA1 condition gives a stronger sensitivity to cisplatin than Δ USP48 Δ FANCC this is actually a meaningless experiment and fails to support the authors' hypothesis.

Minor Points:

(1) The authors find subunits of the BTR complex as suppressors, particularly of Δ FANCD2 and Δ FANCI. It is unclear to me why they specifically mention only BLM and RMI1 in the text, even though Fig. 2a shows TOP3a as well. Could TOP3a be included in Fig. 2b, too. Could the authors speculate, why they see primarily a rescue in Δ FANCD2 and Δ FANCI?

(2) Fig. 3c: The optical impression from the colony forming assay is that Δ USP48 would be hyperresistant to MMC, which is not what the authors conclude in the text. How reproducible is this result.

(3) Fig. 5ab: I would prefer if the control cell lines would be moved from the supplementary to the main figure.

(4) Fig. 5cd: I would prefer if the control cell lines would be moved from the supplementary to the main figure.

(5) discussion: l. 250 "...taken unbiased approaches...", should be "taken an unbiased approach"

Reviewer #2 (Remarks to the Author):

Fanconi anemia (FA) suffers from bone marrow failure due to the defective repair of DNA inter-strand crosslinks (ICLs). In this manuscript, the authors conducted an unbiased screen the genetic synthetic viable interactions for FA deficiency and identified USP48 to be a player that could ameliorate the survival of FA deficiency cells via BRCA1-dependent manner for clearance of DNA damage. While identification of USP48 as an interplay factor with FA is novel, the lack of in-depth mechanism in the present work dampens the enthusiasm and quality for this manuscript. In addition, currently collected data is not enough to support the conclusion. Following are some major concerns:

Following are my comments:

1. The current manuscript suggests that the deubiquitination activity of USP48 is essential for cell survival of FA deficiency cells. Thus, identification of target for USP48 in DNA damage pathway is critical to characterize the deubiquitinating feature and impact for USP48 in association with FA in ICLs. Although several candidate targets were considered, including histone H2A and CtIP, the substrate needs to be confirmed to fulfill the mechanistic study in this manuscript.
2. While depleted of USP48 alleviate the DNA damage hypersensitivity of FA-defective cells, whether add back wildtype or C98S mutant USP48 could restore DNA damage hypersensitivity?
3. DNA repair reporter assays include gene conversion, single-strand annealing and non-homologous end-joining. These assays need to be included to dissect the role of USP48 in the critical step of ICL repair.
4. While the screen is based on all five FA genes: FANCA, FANCC, FANCG, FANCI and FANCD2, in following up experiments, deciphering USP48 was only performed in FANCC genetic background. Genetic screen results indicated that USP48 was also exist in FANCA, FANCG, FANCI and FANCD2 genetic background (Figure 2B). The author need to dissect whether USP48 is only for FANCC cell survival or for all five FA genes background.
5. The author hypothesized that USP48 could function by deubiquitinating the same substrate or substrates that is/are ubiquitylated by E3-ligase BRCA1/BARD1. To assure this hypothesis, the experiment of add back of BRCA1 WT or BRCA1 without E3 ligase activity mutant should be done. Furthermore, the concept about BRCA1 to be an E3 ligase is challenged by the mouse genetic analyses from Baer, R and Ludwig, T. laboratories. Thus, USP48 acts as a deubiquitinating enzymes involving in BRCA1 E3 ligase function needs more discussion.
6. In figure 5C-D, depleted USP48 decrease Rad51 foci in None-MMC treatment cells, this is conflict with figure S4C-D. The conclusion "suggesting that USP48 may promote HR processes even in a DNA repair proficient background" is not solid.
7. In figure 6A-B, the western blot of BRCA1, USP48, et al should be shown after BRCA1 knock down.
8. The author showed USP48 loss enhances recruitment of BRCA1 and RAD51 to DNA damage in FA-deficient cells. Further data showing whether modulating of USP48 expression affect Rad51 foci, r-H2AX foci and HR depend on BRCA1 should be completed.
9. Line 221 "The RAD51 focus formation was also enhanced in the single Δ USP48 cells compared to WT (Supp Fig 4 c-d), suggesting that USP48 may promote HR processes even in a DNA repair proficient background." Should be suggested that USP48 may inhibit HR processes.
10. Fig S1b lacks the WT control. So the conclusion that "we noted that the loss of FANCA affected the protein stability of FANCG and vice-versa. We also observed a reduction in FANCG protein levels in FANCD2-deficient cells as well as a reduction in FANCI protein in FANCD2-deficient cells

(and vice-versa)" is not completely credible.

11. Please make sure the cellular location of USP48? Some paper indicated the USP48 are major localized in cytosol (FASEB J. 2014 Mar; 28(3):1422-34.)

12. USP48 has been reported regulates Mdm2 protein levels (Sci Rep. 2017 Feb 24;7:43180.), while MDM2-P53 activity may attenuate the FA DNA repair pathway (Nat Communs. 2016 Apr 1;7:11091). Necessary connection needs to be discussed.

13. This present work in this manuscript lacks in vivo physiological studies. Currently, there are a lot of FA mice model has been established, such as FancA^{-/-}, FancC^{-/-}, FancG^{-/-}, FancD2^{-/-}, FancD1/Brca2^{-/-} mice (Mutat Res. 2009 July 31; 668(1-2): 133-140.). The author should consider some of these FA models to address the physiological relevance USP48 in association with FA and ICLs.

We thank the reviewers for their comments to improve our manuscript. A point-by-point response to the reviewers' comments is provided below. Statements by the reviewers are given in **bold text** with our response in plain text. **In red are new data/text that have been added to the revised manuscript.**

Reviewer #1 (Remarks to the Author):

This is a short paper describing the identification of genetic interactors with knock-out mutants of human Fanconi Anemia genes. Most intriguingly, the paper describes a genetic suppressor of FA mutant phenotypes – the DUB USP48, as well as providing some details on the mechanism of suppression. This is a tantalizing paper. One immediately thinks of possible next steps – eg what is the USP48 target responsible for suppression or what is the effect of pharmacological inhibition of USP48. Nonetheless, the present finding is in my opinion sufficient in principle for publication in Nature Communications, if the authors address the following points.

We thank the review for these positive remarks.

Major points:

(1) “Recently the concept of synthetic viability (also termed synthetic rescue or genetic suppression) has emerged as a powerful potential approach to ameliorate defects associated with human genetic diseases⁵⁻⁸.” I found this sentence misleading, the concept of genetic suppression is a classical concept in genetics and I would urge the authors to introduce this properly, ie they are using a very classical concept, but putting it to new use.

This has been **re-phrased in the text lines 62-64.**

(2) Fig. 4d is lacking Δ USP48. This would be necessary to judge the suppressive effect in the double ko line. Also, I would find it much more useful if the different cell lines would be plotted in the same graph so that a comparison would be easier.

Fig 4c-d include all four genetic backgrounds.

(3) Fig. 5a-b lack the controls for the shBRCA1 in WT background, as well as in the single mutant backgrounds. This is very important, because if the shBRCA1 condition gives a stronger sensitivity to cisplatin than Δ USP48 Δ FANCC this is actually a meaningless experiment and fails to support the authors' hypothesis.

We presume that the reviewer is referring to Figure 6a-b. Controls **have now been added to Fig 6a-b** and discussed appropriately in the text.

Minor Points:

(1) The authors find subunits of the BTR complex as suppressors, particularly of Δ FANCD2 and Δ FANCI. It is unclear to me why the specifically mention only BLM and RMI1 in the text, even though Fig. 2a shows TOP3a as well. Could TOP3a be included in Fig. 2b, too. Could the authors speculate, why they see primarily a rescue in Δ FANCD2 and Δ FANCI?

The ranking in the plot in Fig2b is a combination of 2 parameters: frequency of hits among different screens and ranking of hits in individual screens based on statistical significance (q-value). TOP3A, although shared between two different screens (FANCI and FANCD2) may have similar ranking with the last gene included in the top10 (RPS25), this is why it was not included in top hits. For further information on the BLM complex synthetic viability we refer the reviewer to a recent publication, which is **now also mentioned in the text (lines 133-135) (Moder M, Velimezi G et al, Nat. Comm. 2017 ref#17)**

(2) Fig. 3c:The optical impression from the colony forming assay is that Δ USP48 would be hyperresistant to MMC, which is not what the authors conclude in the text. How reproducible is this result.

We repeated the experiment in 2 replicates. The present **quantification is a combination of two biological replicates with two technical replicates each**. According to statistical analysis, the small difference in survival between WT and Δ USP48 is not significant, **also commented in the text (lines 158-159)**

(3) Fig. 5ab: I would prefer if the control cell lines would be moved from the supplementary to the main figure.

Figure **5a-b has been re-arranged** as suggested.

(4) Fig. 5cd: I would prefer if the control cell lines would be moved from the supplementary to the main figure.

Figure 5c-d has been **re-arranged as suggested but now appears in Sup Fig 5c-d**. This is because a **new experiment with immunofluorescent staining of RAD51 in U2OS cells is included in the Fig 5c-d** (with all controls) and the previous experiment in HAP1 cells has been moved to the Sup Fig5c-d.

(5) discussion: l. 250 “...taken unbiased approaches...”, should be “taken an unbiased approach”

This has been **corrected, line 329**

Reviewer #2 (Remarks to the Author):

Fanconi anemia (FA) suffers from bone marrow failure due to the defective repair of DNA inter-strand crosslinks (ICLs). In this manuscript, the authors conducted an unbiased screen the genetic synthetic viable interactions for FA deficiency and identified USP48 to be a player that could ameliorate the survival of FA deficiency cells via BRCA1-dependent manner for clearance of DNA damage. While identification of USP48 as an interplay factor with FA is novel, the lack of in-depth mechanism in the present work dampens the enthusiasm and quality for this manuscript. In addition, currently collected data is not enough to support the conclusion. Following are some major concerns:

We thank the reviewer for the encouraging remarks on novelty of USP48 and find that by having addressed the concerns of both reviewers, we now have a better mechanistic understanding of the role of USP48 in the repair of ICLs.

Following are my comments:

1. The current manuscript suggests that the deubiquitination activity of USP48 is essential for cell survival of FA deficiency cells. Thus, identification of target for USP48 in DNA damage pathway is critical to characterize the deubiquitinating feature and impact for USP48 in association with FA in ICLs. Although several candidate targets were considered, including histone H2A and CtIP, the substrate needs to be confirmed to fulfill the mechanistic study in this manuscript.

We have tested RPA70 and H2A as potential substrates of USP48. We observe a difference in H2A ubiquitylation levels but not in ubiquitylation levels of RPA70 (**Fig 6c and Sup Fig 5h, respectively**).

2. While depleted of USP48 alleviate the DNA damage hypersensitivity of FA-defective cells, whether add back wildtype or C98S mutant USP48 could restore DNA damage hypersensitivity?

Unfortunately the reconstitution experiments to alleviate the DNA damage hypersensitivity did not work, technically. Whereas we could express the USP48 constructs in WT cells and in Δ FANCC cells, we could not express the USP48 C98S mutant in Δ FANCC Δ USP48 cells and we could only partially express WT USP48 (see Fig 1, below). Hence, the WT construct only shows partial resensitization. We do however present other data in our manuscript that suggests that USP48 activity is implicated:

- The C98S mutant shows longer recruitment at sites of damage (Supp Fig 4b-c).
- H2A ubiquitin levels partially depend on presence of USP48 (Fig 6c).

Figure 1. Inducible expression of WT and C98S USP48 in WT, Δ FANCC, Δ USP48 Δ FANCC HAP1 cells. Stable cell lines carrying an inducible USP48 (WT or C98S) under a Tet-regulated promoter were generated. (a) Western blot showing expression levels of USP48 (WT and C98S) after induction with Doxycycline (2 μ g/ml) for 48 hours (New doxycycline was added after the first 24h) or no induction (-). (b) Colony formation with cells shown in (a). Induced cells were seeded in Doxycycline-containing medium and 24h later MMC was added at the indicated doses and new doxycycline for the induced cells. Medium with fresh Doxycycline was replenished every 2 days. Cells were fixed 6 days after treatment with MMC.

3. DNA repair reporter assays include gene conversion, single-strand annealing and non-homologous end-joining. These assays need to be included to dissect the role of USP48 in the critical step of ICL repair.

We performed the TLR assay to measure HR-dependent repair of I-SceI-induced DNA double-strand breaks, although the lesion is different form ICL (Sup Fig 5e). We do not see a role for USP48 in the repair of such lesions. This point has been discussed in the text.

4. While the screen is based on all five FA genes: FANCA, FANCC, FANCG, FANCI and FANCD2, in following up experiments, deciphering USP48 was only performed in FANCC genetic background. Genetic screen results indicated that USP48 was also exist in FANCA, FANCG, FANCI and FANCD2 genetic background (Figure 2B). The author need to dissect whether USP48 is only for FANCC cell survival or for all five FA genes background.

We performed experiments with $\Delta FANCG$ and $\Delta FANCD2$ cells where we observed that loss of USP48 (in a polyclonal population of deficient FA cells for USP48, generated by CRISPR-Cas9) alleviates MMC-included toxicity. Results are included in Supp Fig 2e-f.

5. The author hypothesized that USP48 could function by deubiquitinating the same substrate or substrates that is/are ubiquitylated by E3-ligase BRCA1/BARD1. To assure this hypothesis, the experiment of add back of BRCA1 WT or BRCA1 without E3 ligase activity mutant should be done. Furthermore, the concept about BRCA1 to be an E3 ligase is challenged by the mouse genetic analyses from Baer, R and Ludwig, T. laboratories. Thus, USP48 acts as a deubiquitinating enzymes involving in BRCA1 E3 ligase function needs more discussion.

We performed several experiments as suggested by the reviewer but unfortunately we faced technical difficulties and below we show two such experiments and explain the difficulties we have faced.

In the first experiment, siRNA-mediated depletion of BARD1 (an essential gene) resulted in loss of cellular viability, even in the absence of MMC (Fig 2a-b, below). This loss of viability was even more pronounced in cells lacking FANCC, with hardly any cells surviving on which to perform clonogenic experiments on.

In the second experiment, where we achieved a less successful depletion of BARD1, this did not result in any cellular sensitivity to MMC (Fig 2c-d, below) hence there was no sensitivity to reconstitute with either WT or catalytically inactive BARD1.

Figure 2. Dependency of the synthetic rescue interaction between USP48 and FANCC, on BARD1 activity. (a) Immunoblot for BARD1 and actin for the indicated cell lines transfected with siRNA targeting BARD1. (b) Colony formation of the indicated cell lines after MMC treatment for 7 days. (c) Immunoblot to assess the over-expression of BARD1 WT and BARD1 R99E in the indicated cell lines after transfection with siRNA targeting BARD1. (d) Colony formation of the indicated cell lines after MMC treatment for 7 days.

6. In figure 5C-D, depleted USP48 decrease Rad51 foci in None-MMC treatment cells, this is conflict with figure S4C-D. The conclusion “suggesting that USP48 may promote HR processes even in a DNA repair proficient background” is not solid.

The effect of USP48 loss on RAD51 foci refers to repair of lesions upon MMC, ie ICLs. The effect of USP48 loss or depletion on basal RAD51 foci in Δ FANCC cells could be due to reduced endogenous damage arising from replication stress in these cells due to loss of FANCC. Experiments in U2OS (new Fig 5c-d) shows that USP48 loss affects RAD51 foci formation upon MMC mainly in the Δ FANCC background. **The conclusion has been re-phrased to better match the results (line 248)**

7. In figure 6A-B, the western blot of BRCA1, USP48, et al should be shown after BRCA1 knock down.

Immunoblots for BRCA1, USP48 and FANCC have been added in Fig 6a.

8. The author showed USP48 loss enhances recruitment of BRCA1 and RAD51 to DNA damage in FA-deficient cells. Further data showing whether modulating of USP48 expression affect Rad51 foci, r-H2AX foci and HR depend on BRCA1 should be completed

We repeated the immunofluorescent staining for γ H2AX in shBRCA1 cells. However, we observe increased toxicity upon BRCA1 depletion (an essential gene) that hinders the experiment, since many cells die by the time the foci are measured (this could explain the lower levels of γ H2AX in the Δ FANCC shBRCA1 cells). We do see a significance difference between EV and shBRCA1 in the Δ FANCC Δ USP48 cells, but we feel that technical limitations when working with essential genes, such as BRCA1 prevent the extraction of safe conclusions from these experiments (see Fig 3, below).

Figure 3. Immunofluorescence for γ H2AX with shBRCA1 cells. (a) Representative immunofluorescence images for γ H2AX foci and DAPI staining. (b) Quantification of γ H2AX foci in the respective cell lines after MMC treatment and recovery with or without BRCA1 depletion. Error bars show mean with SD. Statistical significance was determined by Wilcoxon test with $p < 0.05$ as threshold. UT=Untreated. EV=Empty vector control. Sh=short hairpin RNA. 0h=MMC 50nM for 24 hours.

9. Line 221 “The RAD51 focus formation was also enhanced in the single Δ USP48 cells compared to WT (Supp Fig 4 c-d), suggesting that USP48 may promote HR processes even in a DNA repair proficient background.” Should be suggested that USP48 may inhibit HR processes.

This has been corrected in the text (line 248)

10. Fig S1b lacks the WT control. So the conclusion that “we noted that the loss of FANCA affected the protein stability of FANCG and vice-versa. We also observed a reduction in FANCG protein levels in FANCD2-deficient cells as well as a reduction in FANCI protein in FANCD2-deficient cells (and vice-versa)” is not completely credible.

These blots have been re-run and the WT has been included (Supp Fig 1b). Also, the text referring to this figure has been re-phrased to better match the observed result (lines 93-95)

11. Please make sure the cellular location of USP48? Some paper indicated the USP48 are major localized in cytosol (FASEB J. 2014 Mar; 28(3):1422-34.)

Expression of GFP-tagged USP48 (both WT and catalytically inactive) shows it to be nuclear (Supp Fig 3).

12. USP48 has been reported regulates Mdm2 protein levels (Sci Rep. 2017 Feb 24;7:43180.), while MDM2-P53 activity may attenuate the FA DNA repair pathway (Nat Commun. 2016 Apr 1;7:11091). Necessary connection needs to be discussed.

This has been discussed in the text (lines 375-383)

13. This present work in this manuscript lacks in vivo physiological studies. Currently, there are a lot of FA mice model has been established, such as FancA^{-/-}, FancC^{-/-}, FancG^{-/-}, FancD2^{-/-}, FancD1/Brca2^{-/-} mice (Mutat Res. 2009 July 31; 668(1-2): 133–140.). The author should consider some of these FA models to address the physiological relevance USP48 in association with FA and ICLs.

We have added to the discussion the use of mouse models to elucidate this genetic interaction *in vivo*. (Lines 387-390) However, this work would take a considerable amount of time to setup and hence is beyond the scope of the current study.

Reviewers' comments:

Reviewer #2 (Remarks to the Author):

Loizou et al responded my concerns point-by-point. The authors have done some suggested experiments. While these results improve the manuscript in certain way, there are still questions remained as follows:

1. In the new incorporated fig 6C, in comparison to USP48 knockout (Δ USP48), the wildtype USP48 (WT) cells has much more H2A ubiquitination signals (FK2). The conclusion "increase in the poly-ubiquitinated forms of H2A both in the single Δ USP48 and double Δ USP48 Δ FANCC cells" is not correct. Please also interprets why USP48 (WT) cells has much more H2A ubiquitination signals (FK2).
2. The reconstitution experiments of wildtype or C98S mutant USP48 in USP48 depleted cells was failed due to the author's failure or only partial expression of the USP48 C98S mutant and wildtype in Δ FANCC- Δ USP48 cells. This indicate the doxycycline inducible expression has caveats, therefore the author should try different systems.
3. The author shows that knockdown USP48 inhibits homologous recombination (Sup Fig 5e). As the author indicated that USP48 loss enhances recruitment of BRCA1 and RAD51, these results are not consistent.
4. The author failed to have siRNA-mediated depletion of BARD1 in cells, why the author did not try adding back of BRCA1 WT or BRCA1 without E3 ligase activity mutant as we suggested?
5. To satisfy the minimum level for Nature communication, a piece of in vivo physiological studies is required.

We are pleased that Reviewer #1 was satisfied with our revised manuscript, and we thank Reviewer #2 for his/her comments that aim to further improve our manuscript. A point-by-point response to Reviewer #2 is given below. Statements by the Reviewer are given in **bold text** with our response in plain text. In the text of the manuscript, **in red are previous alterations to the text while in blue are the most recent changes to the text.**

Reviewer #2 (Remarks to the Author):

Loizou et al responded my concerns point-by-point. The authors have done some suggested experiments. While these results improve the manuscript in certain way, there are still questions remained as follows:

1. In the new incorporated fig 6C, in comparison to USP48 knockout (Δ USP48), the wildtype USP48 (WT) cells has much more H2A ubiquitination signals (FK2). The conclusion “increase in the poly-ubiquitinated forms of H2A both in the single Δ USP48 and double Δ USP48 Δ FANCC cells” is not correct. Please also interprets why USP48 (WT) cells has much more H2A ubiquitination signals (FK2).

FK2 antibody detects both mono- and poly-ubiquitin chains. Consequently, we hypothesize that the lower band (to which the reviewer refers) detected by the FK2 antibody corresponds to mono-ubiquitylated H2A. Indeed, incubation with anti-FLAG antibody reveals an even lower band, which we assume corresponds to non-ubiquitylated H2A. Hence, although this lower band of the FK2 blot (mono-ub H2A) appears more abundant in WT cell extract, there is a slight increase in the higher band in the Δ USP48 cell extract, which suggests that there is an increase, although to a much lesser extent than in the Δ USP48 Δ FANCC sample, in the poly-ubiquitin chains of H2A. **This has now been clearly labeled in Figure 6C.**

Regarding the stronger increase observed in the lower FK2-detected band, we see that it is accompanied by an increase in the FLAG-detected bands as well, both in the pulldown and the lysate. This indicates that there is a general higher level of histone FLAG-H2A expression in the WT, for which we do not know the reason but a possibility is differences in plasmid uptake between the cell lines.

2. The reconstitution experiments of wildtype or C98S mutant USP48 in USP48 depleted cells was failed due to the author’s failure or only partial expression of the USP48 C98S mutant and wildtype in Δ FANCC- Δ USP48 cells. This indicate the doxycycline inducible expression has caveats, therefore the author should try different systems.

We have taken various approaches to address this comment as documented below. In summary, we provide four sets of data that support the hypothesis that the catalytic activity of USP48 is important for its DDR functions, and specifically for its genetic interaction with FANCC:

1. We have shown that overexpression of WT USP48 can partially re-sensitize Δ USP48 Δ FANCC cells to MMC while the C98S mutant cannot – **Supp Figure 2c-d; lines 165-169 (new data).** For this experiment, we used transient expression of GFP-tagged USP48 WT and C98S proteins. Cells were seeded 24 h after transfection and MMC was added the following day. While we can confirm that exogenous USP48 levels stay high at the time of MMC addition, it is possible that the protein levels start to decline afterwards, which could account for the weak effect on the survival phenotype, which is observed 6 days post MMC addition.

2. We document that catalytic inactive USP48 localises to sites of ICLs in a manner that is much stronger than that of the WT protein – **Figure 4b and Supp Figure 4b-e.**

3. We show that the C98S catalytic mutant but not WT USP48 colocalises with BRCA1– **Figure 5c; lines 252-260 (new data)**. This experiment was performed by transient expression of GFP-tagged USP48 WT and C98S constructs.

4. In addition, we show that the catalytic mutant but not WT USP48 colocalises with γ H2AX – **Supp Figure 3; lines 196-200 (new data)** by transient expression of GFP-tagged USP48 WT and C98S constructs.

4. The author shows that knockdown USP48 inhibits homologous recombination (Sup Fig 5e). As the author indicated that USP48 loss enhances recruitment of BRCA1 and RAD51, these results are not consistent.

We appreciate this comment from the reviewer since, even though this point was addressed in the **previous version of our manuscript**, it appears not to be clear. For this purpose, **we have further clarified the reasons for which this result is not necessarily inconsistent**. The repair of an I-Sce1 induced DNA double strand break is not directly comparable to the repair of ICLs which was the type of damage that was used in the RAD51 IF experiment. Therefore, the role of USP48 might differ according to the type of lesions induced in the DNA (**lines 277-283**).

5. The author failed to have siRNA-mediated depletion of BARD1 in cells, why the author did not try adding back of BRCA1 WT or BRCA1 without E3 ligase activity mutant as we suggested?

We have performed an experiment where we have re-expressed either wild-type or E3 ligase inactive BRCA1 in Δ USP48 Δ FANCC cells siRNA-depleted for BRCA1– **Supp Figure 5f-g; lines 305-312 (new data)**. Following MMC exposure, we observe that cells expressing wild-type BRCA1, as well as the I26A mutant BRCA1, are more resistant than controls. This indicates that the E3 ligase activity is dispensable for this function in the context of cells deficient in both FANCC and USP48. **We discuss these findings in our revised text (lines 382-398)**.

We thank the reviewer for this comment since it has allowed us to revise our model.

6. To satisfy the minimum level for Nature communication, a piece of in vivo physiological studies is required.

We agree with the reviewer, that *in vivo* work would provide an additional set of interesting and novel data. However, we disagree that this data is normally required to warrant publication of a molecular biology manuscript in *Nature Communications*, and hence we suggest that such studies will be the focus of future research by others and us.